# LRANKER: LLM RANKER FOR MASSIVE CANDIDATES

## ABSTRACT

Large language models (LLMs) have recently shown strong potential for ranking by capturing semantic relevance and adapting across diverse domains, yet existing methods remain constrained by limited context length and high computational costs, restricting their applicability to real-world scenarios where candidate pools often scale to millions. To address this challenge, we propose `LRanker`, a framework tailored for large-candidate ranking. `LRanker` incorporates a candidate aggregation encoder that leverages $K$-means clustering to explicitly model global candidate information, and a graph-based test-time scaling mechanism that partitions candidates into subsets, generates multiple query embeddings, and integrates them through an ensemble procedure. By aggregating diverse embeddings instead of relying on a single representation, this mechanism enhances robustness and expressiveness, leading to more accurate ranking over massive candidate pools. We evaluate `LRanker` on seven tasks across three scenarios in *RBench* with different candidate scales. Experimental results show that `LRanker` achieves over *30%* gains in the *RBench-Small* scenario, improves by *3–9%* in MRR in the *RBench-Large* scenario, and sustains scalability with *20–30%* improvements in the *RBench-Ultra* scenario with more than *6.8M* candidates. Ablation studies further verify the effectiveness of its key components. Together, these findings demonstrate the robustness, scalability, and effectiveness of `LRanker` for massive-candidate ranking.

## 1 INTRODUCTION

Using large language models (LLMs) for ranking has already demonstrated remarkable potential (Li et al., 2023b; Lin et al., 2024; Jiang et al., 2023), showing strong capabilities in capturing semantic relevance, adapting to diverse domains, and achieving competitive performance compared to traditional retrieval and ranking methods. However, constraints such as limited context length (Rashid et al., 2024; Liu et al., 2024b) and prohibitive computational costs (Chen et al., 2025b) restrict current LLM-based ranking methods to small candidate sets, limiting their applicability to real-world scenarios like search and recommendation, where candidate pools often scale to millions. Therefore, our paper aims to raise attention to this pressing research question: *How can we build an efficient LLM ranker for large candidate ranking?*

Existing LLM-based rankers can be broadly distinguished by their input and output formats as shown in Table 1. In terms of input, prior approaches typically adopt one of four strategies: (1) query only (Li et al., 2023a), (2) query combined with a single candidate (Ma et al., 2024), (3) query–candidate pairs (Qin et al., 2023), or (4) the full candidate list (Pradeep et al., 2023; Feng et al., 2025; Sun et al., 2023a). While the last option quickly becomes infeasible due to the limited context length of LLMs, the first three fail to incorporate global candidate-level information, introducing systematic biases into the ranking process. On the output side, most methods directly generate ranking results in the token space, which couples ranking quality with the LLM's decoding latency and restricts scalability.

Based on the above discussion, we argue that *an effective LLM framework for massive-candidate ranking must model global candidate information in the input and perform ranking through embedding-based outputs.* Nevertheless, constructing such LLM rankers faces two key challenges. First, when the number of candidates is large, the limited context length of LLMs makes it difficult to model the global candidate information, which can lead to ranking inaccuracies. Second, relying on a single embedding to rank all candidates limits the expressive capacity of the model, thereby constraining its overall potential.

Table 1: **Comparison of `LRanker` with existing LLM-based rankers across four dimensions: input, output, ranking latency, and maximum candidate scale.** Unlike prior approaches, `LRanker` leverages aggregated candidate centroids within an efficient LLM-based ranking architecture, making its computation independent of candidate size and enabling efficient processing of the information of large-scale candidate sets.

| LLM-based Ranker | Input | Output | Ranking Latency | Largest Candidate Scale |
|---|---|---|---|---|
| PRP (Qin et al., 2023) | Query+ Candidate Pair | Token | High | 100 |
| RankGPT (Sun et al., 2023a) | Query + Candidate List | Token | Moderate | 100 |
| IRanker (Feng et al., 2025) | Query + Partial Candidate List | Token | Moderate | 20 |
| RankLLaMA (Ma et al., 2024) | Query+Single Candidate | Embedding | High | 200 |
| `LRanker` | Query + Aggregated Candidate info | Embedding | **Low** | **6.81M** |

To address the limitations of existing LLM rankers, we propose `LRanker`, a framework tailored for large-candidate ranking. At the input stage, `LRanker` employs a candidate aggregation encoder that clusters candidate embeddings via $K$-means and summarizes them into compact centroids, ensuring that global candidate information is explicitly modeled within the prompt. At the inference stage, `LRanker` introduces a graph-based test-time scaling mechanism that iteratively partitions candidates, generates multiple query embeddings under different candidate subsets, and integrates them through an ensemble procedure. This design enriches the representation of the query by aggregating multiple perspectives rather than relying on a single embedding, thereby enhancing robustness and discriminative power for ranking, and enabling more accurate matching across massive candidate pools.

We evaluate `LRanker` on seven tasks across three scenarios in *RBench* with different candidate scales. In the *RBench-Small* setting, `LRanker` achieves over *30%* relative gains compared with existing rankers. In the *RBench-Large* setting, it outperforms existing approaches by about *3–9%* in MRR. Even in the challenging *RBench-Ultra* scenario with more than *6.8M* candidates, `LRanker` sustains scalability and delivers *20–30%* improvements. Ablation studies further confirm that global candidate aggregation, test-time ensemble, and LoRA adaptation all contribute to these gains, demonstrating the robustness of our design.

## 2 PROBLEM FORMULATION

Given a query $q$, the objective of a ranking task (Liu et al., 2009; Li, 2011; Cao et al., 2007) is to train a ranker $f$ that orders a candidate set $D = \{c_1, c_2, \ldots, c_n\}$ of size $n$. Typically, $D$ can be separated into a positive subset $D_p$ (items that the user truly interacted with, e.g., products actually purchased) and a negative subset $D_n$ (items not chosen). To assess how accurately the ranker retrieves the positives, its performance is evaluated with ranking metrics $E$, such as Normalized Discounted Cumulative Gain (nDCG) (Järvelin & Kekäläinen, 2002) or Mean Reciprocal Rank (MRR) (Voorhees et al., 1999; Cremonesi et al., 2010).

Formally, a ranker $\pi$ maps the pair $(q, D)$ into an ordered sequence

$$\pi : (q, D) \mapsto O = \{c_1^{r_1}, c_2^{r_2}, \ldots, c_n^{r_n}\}, \quad O \in \mathbb{S}_n, \tag{1}$$

where $r_i$ denotes the position assigned to candidate $c_i$, and $\mathbb{S}_n$ is the space of all permutations over $n$ elements. The learning objective is then to identify the optimal ranker $\pi^*$ within a hypothesis class $\mathcal{F}$ that maximizes the expected evaluation score under the data distribution $\mathcal{Z}$:

$$\pi^* = \arg\max_{f \in \mathcal{F}} \; \mathbb{E}_{(q,D)\sim\mathcal{Z}} \left[ E(\pi(q, D)) \right]. \tag{2}$$

## 3 LRANKER: LLM RANKER FOR MASSIVE CANDIDATES

Building on the limitations of existing LLM rankers summarized in Figure 1, we introduce `LRanker`, an encoder–decoder framework designed to handle large-candidate ranking more effectively. `LRanker` improves ranking performance through two complementary components. First, a *candidate aggregation encoder* applies $K$-means clustering to offline candidate embeddings and uses a learnable projector to inject the resulting aggregated vector into the LLM as a soft prompt, enabling the model to condition on global candidate information when forming query and candidate representations (Figure 1(b)). Second, a *graph-based test-time scaling* strategy iteratively partitions and prunes the candidate set, recomputes partition-specific query embeddings, and aggregates them

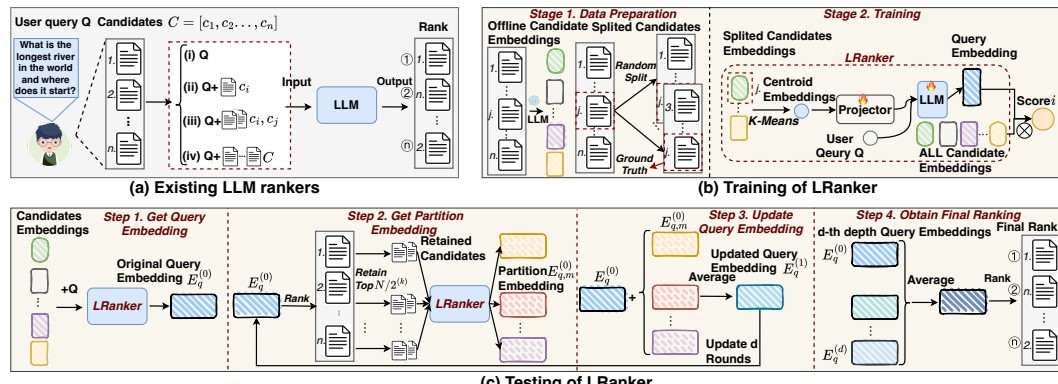

Figure 1: **Compared with existing LLM rankers on large-candidate tasks, `LRanker` incorporates advanced designs in both the representation of candidate information and the inference strategies used during testing. Note that the spark icon denotes models that require fine-tuning, while the snowflake icon denotes models with frozen weights.** (a) Existing LLM rankers generally adopt four input formats (highlighted in the red box): (i) query only, (ii) query with a single candidate, (iii) query with candidate pairs, and (iv) query with the complete candidate list. The (iv) setting is fundamentally constrained by the limited context length of LLMs in massive-candidate scenarios, while the first three fail to incorporate global candidate-level information, leading to systematic biases in ranking. (b) `LRanker`'s training pipeline. In Stage 1 (data preparation), offline candidate embeddings are randomly split and clustered by $K$-means to obtain centroid embeddings. In Stage 2 (training), these centroids are passed through the learnable projector and injected into the LLM decoder via a special placeholder token, yielding query embeddings that already condition on global candidate information and are optimized with a ranking loss. (c) Graph-based test-time scaling. Step 1 obtains an initial query embedding $E_q^{(0)}$ on the full candidate set. Step 2 uses $E_q^{(0)}$ to retain top-ranked candidates within each partition and recompute partition-specific embeddings. Step 3 updates the query embedding by averaging embeddings from multiple partitions and repeating this elimination-and-update process for $d$ rounds, producing $\{E_q^{(t)}\}_{t=0}^d$. Step 4 averages the scores computed from these embeddings to obtain the final ranking, enabling robust performance across diverse candidate scales.

across multiple candidate scales to enhance inference robustness (Figure 1(c)). We also describe the training procedure, including random partition sampling for robust query representations, and provide a motivating example to illustrate how these components work together.

## 3.1 FRAMEWORK OF LRANKER

We design `LRanker` as an encoder–decoder framework consisting of two trainable components: a *candidate aggregation encoder* and an *LLM decoder*. The overall goal is to learn query-aware embeddings that integrate both user intent and global candidate information, thereby enabling more accurate passage ranking. Formally, let $\mathcal{P}_\phi$ denote the projection network (with parameters $\phi$) and let $f_\theta$ denote the LLM decoder (with parameters $\theta$). Both $\phi$ and $\theta$ are optimized under the ranking loss described in Section 3.2. The overall pipeline corresponds to Stages 1 and 2 in Figure 1(b).

**Candidate Aggregation Encoder**. Following Stage 1 (Data Preparation) in Figure 1(b), we first construct a compact representation of the global candidate set. Given a candidate set $\mathcal{C} = \{c_1, c_2, \ldots, c_N\}$ of size $N$, we obtain offline base encoder embeddings $\mathbf{e}(c_i)$ for each candidate $c_i$ and cache them for reuse during training and testing. To capture the global distributional structure of candidates, we then apply $K$-means clustering on these embeddings:

$$\{\mathcal{G}_1, \mathcal{G}_2, \ldots, \mathcal{G}_K\} = \text{KMeans}(\{\mathbf{e}(c_1), \ldots, \mathbf{e}(c_N)\}), \tag{3}$$

where $\mathcal{G}_k$ represents the $k$-th cluster. Each cluster centroid is computed as

$$\mathbf{g}_k = \frac{1}{|\mathcal{G}_k|} \sum_{c_i \in \mathcal{G}_k} \mathbf{e}(c_i). \tag{4}$$

The $K$ centroids are concatenated and then projected into the LLM embedding space by a learnable projector:

$$\mathbf{g} = [\mathbf{g}_1; \mathbf{g}_2; \ldots; \mathbf{g}_K], \quad \tilde{\mathbf{g}} = \mathcal{P}_\phi(\mathbf{g}), \tag{5}$$

where $\mathcal{P}_\phi$ is implemented as a small MLP whose parameters $\phi$ are trained jointly with the LLM parameters $\theta$. In Figure 1(b), this corresponds to the red-boxed "Centroid Embeddings" and "Projector" modules that transform offline candidate embeddings into a single aggregated vector.

**LLM as a Decoder.** The LLM $f_\theta$ serves as a decoder that jointly encodes the query and the aggregated candidate information. We design an input prompt that integrates the user query $q$ with the projected aggregated candidate embedding $\tilde{\mathbf{g}}$, as shown in Appendix A and Stage 2 of Figure 1(b). Concretely, we construct a discrete token sequence the in-context input $(x_1, x_2, \ldots, x_L)$ that includes system instructions, the textual query $q$, and a special placeholder token <|embedding|> at a fixed position $p$. Let $E_{\text{tok}}$ be the LLM token embedding matrix. We first map all tokens to embeddings and then *replace* the embedding at position $p$ by the continuous vector $\tilde{\mathbf{g}}$:

$$\mathbf{x}_t = \begin{cases} E_{\text{tok}}(x_t), & t \neq p, \\ \tilde{\mathbf{g}}, & t = p. \end{cases} \tag{6}$$

The resulting sequence of input embeddings $\mathbf{X}_0 = (\mathbf{x}_1, \ldots, \mathbf{x}_L)$ is then fed into the LLM, producing the hidden representations $(\mathbf{z}_1, \ldots, \mathbf{z}_L) = f_\theta(\mathbf{X}_0)$, where $\mathbf{z}_t$ denotes the final-layer hidden state at position $t$. Operationally, this is equivalent to using a single-token *soft prompt*: the placeholder position is occupied by a continuous vector $\tilde{\mathbf{g}}$ rather than a discrete token embedding, and gradients flow back to both $\mathcal{P}_\phi$ and $f_\theta$. No additional modifications to intermediate layers are required.

To define a query embedding, we explicitly append an end-of-sequence token <eos> to the prompt. For clarity, we use <eos> as a generic notation for the model-specific end-of-text token (e.g., the end-of-sequence token in Qwen-3). For a query $q$ with $T_q$ textual tokens, the full sequence thus contains the $T_q$ query tokens plus the <eos> token. Let $\mathbf{z}_{q,t}$ denote the hidden state of the $t$-th query token and $\mathbf{z}_{q,\text{eos}}$ the hidden state at the <eos> position returned by $f_\theta$. Because the <eos> position is used by the LLM to predict the next token, we refer to $\mathbf{z}_{q,\text{eos}}$ as the "next-token" hidden state in the sequel. The final query embedding is obtained by averaging over all query-token hidden states and this next-token state:

$$\mathbf{h}_q = \frac{1}{T_q + 1} \left( \mathbf{z}_{q,\text{eos}} + \sum_{t=1}^{T_q} \mathbf{z}_{q,t} \right). \tag{7}$$

Similarly, for each candidate $c_i$ with length $|c_i|$, we obtain an off-line candidate embedding by feeding its text (plus an <eos> token) into the same LLM $f_\theta$.[1] Let $\mathbf{z}_{c_i,j}$ and $\mathbf{z}_{c_i,\text{eos}}$ denote the hidden states of the $j$-th candidate token and the <eos> position, respectively. We define

$$\mathbf{h}_{c_i} = \frac{1}{|c_i| + 1} \left( \mathbf{z}_{c_i,\text{eos}} + \sum_{j=1}^{|c_i|} \mathbf{z}_{c_i,j} \right). \tag{8}$$

Note that $\mathbf{h}_q$ and $\mathbf{h}_{c_i}$ are produced by the same LLM $f_\theta$, with the only difference that $\mathbf{h}_q$ additionally conditions on the injected global candidate vector $\tilde{\mathbf{g}}$ at the placeholder position.

Finally, we compute relevance scores and rank the candidates by inner product:

$$s(q, c_i) = \langle \mathbf{h}_q, \mathbf{h}_{c_i} \rangle, \quad \pi(q) = \text{argsort}\left( \{s(q, c_i)\}_{i=1}^N \right), \tag{9}$$

where $\langle \cdot, \cdot \rangle$ denotes the inner product and $\pi(q)$ represents the ordered sequence of candidates. This encoder–decoder design (visualized in Figure 1(b)) makes it explicit that both the projector $\mathcal{P}_\phi$ and the LLM $f_\theta$ are trainable, and clarifies how the continuous aggregated embedding $\tilde{\mathbf{g}}$ is injected into the discrete token sequence to obtain query and candidate representations.

## 3.2 TRAINING LRANKER

The overall training pipeline in Figure 1(b) contains two stages. Stage 1 (Data Preparation) constructs offline candidate embeddings and their $K$-means centroids, while Stage 2 (Training) feeds the aggregated vector into the LLM decoder together with the user query. During this stage, we optimize all trainable parameters $\Theta = \{\theta, \phi\}$ of the LLM decoder $f_\theta$ and the projector $\mathcal{P}_\phi$ under a ranking objective with both positive and negative samples. For each query $q$, let $c^+$ denote the ground-truth

---

[1]For efficiency, this step can be precomputed and cached.

Table 2: **Detailed summarization of tasks used in our ranking experiments with varying candidate scales.** We summarize the scenarios, task names, candidate sizes, and the number of queries.

| Scenario | Task | Candidate Size | # Query Num |
|---|---|---|---|
| **RBench-Small** | | | |
| Rec-Music | Recommendation | 20 | 12,483 |
| Routing-Balance | Routing | 20 | 1,620 |
| **RBench-Large** | | | |
| Rec-Movie | Recommendation | 3,884 | 4,167 |
| Rec-Toy | Recommendation | 11,925 | 19,413 |
| Rec-Video | Recommendation | 25,600 | 94,800 |
| Rec-Software | Recommendation | 17,600 | 146,400 |
| MS MARCO | Passage ranking | 24,697 | 3,038 |
| ESCI | Product searching | 4,000 | 3,999 |
| **RBench-Ultra** | | | |
| Rec-Clothing | Recommendation | 6,805,462 | 185,925 |

relevant candidate and $\mathcal{C}^- = \{c_1^-, \ldots, c_M^-\}$ the set of sampled negative candidates. Given the relevance score $s(q, c; \Theta)$ defined in Eq. (7), the model is trained to assign a higher score to $c^+$ than to any negative candidate $c^- \in \mathcal{C}^-$. Concretely, the loss function is defined as a softmax cross-entropy:

$$\mathcal{L}(q, c^+, \mathcal{C}^-; \Theta) = -\log \frac{\exp(s(q, c^+; \Theta))}{\exp(s(q, c^+; \Theta)) + \sum_{c^- \in \mathcal{C}^-} \exp(s(q, c^-; \Theta))}, \quad (10)$$

where $s(q, c; \Theta)$ is the inner-product score between the query and candidate embeddings produced by `LRanker`.

**Random partition sampling.** In Figure 1(b), the "Random Split" module reflects a key design for robust training: *random partition sampling*. To improve the robustness of the aggregated candidate representation and prepare the model for test-time scaling, we introduce this strategy during training. For each training instance with candidate set $\mathcal{C} = \{c_1, \ldots, c_N\}$, we first randomly split $\mathcal{C}$ into $M$ disjoint subsets:

$$\mathcal{C} = \mathcal{C}^{(1)} \cup \cdots \cup \mathcal{C}^{(M)}, \quad \mathcal{C}^{(m)} \cap \mathcal{C}^{(m')} = \emptyset \text{ for } m \neq m'. \quad (11)$$

At each optimization step, we then uniformly sample one index $r \in \{1, \ldots, M\}$ and compute the aggregated candidate vector

$$\tilde{\mathbf{g}}^{(r)} = \mathcal{P}_\phi\big(\text{Aggregate}(\mathcal{C}^{(r)})\big), \quad (12)$$

where $\text{Aggregate}(\cdot)$ denotes the $K$-means-based centroid extraction described in Section 3.1. The sampled vector $\tilde{\mathbf{g}}^{(r)}$ is injected into the prompt via the special placeholder token (Eqs. (4)–(5)), and the LLM decoder $f_\theta$ produces the query embedding $\mathbf{h}_q^{(r)}$ and candidate embeddings. These embeddings are then used to compute scores $s(q, c; \Theta)$ and update the parameters via the loss $\mathcal{L}(q, c^+, \mathcal{C}^-; \Theta)$.

This random partition sampling serves as a form of data augmentation over candidate contexts: the query representation is trained to be stable with respect to different subsets of candidates and varying candidate scales. As a result, the "Query Embedding" output in Figure 1(b) is already robust to changes in the candidate context. Consequently, at test time, `LRanker` can naturally leverage multiple candidate partitions of different sizes and aggregate their contributions for improved ranking performance, as discussed in Section 3.3.

### 3.3 GRAPH-BASED TEST-TIME SCALING

Motivated by the principle of ensemble learning (Zhou, 2025; Breiman, 1996; Freund & Schapire, 1997; Wolpert, 1992)—where combining multiple classifiers outperforms relying on a single one—we extend this idea to ranking by aggregating multiple query embeddings produced under different candidate partitions. The procedure is summarized in Figure 1(c), which decomposes our test-time strategy into four steps (Step 1–Step 4). The key intuition is that a single query embedding may be biased by the initial candidate context, while combining embeddings obtained from diverse partitions can lead to more robust ranking performance.

Concretely, for a ranking task with $N$ candidates, we first obtain an initial query embedding $E_q^{(0)}$ using the proposed encoder–decoder framework applied to the full candidate set (Step 1 in Figure 1(c)). Based on $E_q^{(0)}$, we perform an *elimination* step (Step 2): the candidates are partitioned into $k$ disjoint subsets of size approximately $N/k$, and within each subset we retain only the top-ranked candidates according to $E_q^{(0)}$. This yields a reduced candidate pool $\mathcal{C}^{(1)}$ that is still diverse but more focused on promising items.

For each of the $k$ subsets in this reduced pool, we recompute the aggregated candidate information (via the same $K$-means-based encoder in Section 3.1) and obtain a new query embedding conditioned on that subset (Step 3). This yields $k$ embeddings, which are then averaged with the original $E_q^{(0)}$ to produce an updated embedding $E_q^{(1)}$:

$$E_q^{(1)} = \frac{1}{k+1} \left( E_q^{(0)} + \sum_{m=1}^{k} E_{q,m}^{(0)} \right), \tag{13}$$

where $E_{q,m}^{(0)}$ denotes the embedding obtained from the $m$-th partition in the first elimination round. Intuitively, $E_q^{(1)}$ aggregates information from multiple "views" of the candidate set.

This elimination-and-update process can be repeated for multiple iterations, producing a sequence of embeddings $E_q^{(0)}, E_q^{(1)}, \dots, E_q^{(d)}$. At test time (Step 4), the final ranking score for a candidate $c$ is computed by averaging its scores across all embeddings in the sequence:

$$s_{\text{final}}(q, c) = \frac{1}{d+1} \sum_{t=0}^{d} s_{E_q^{(t)}}(q, c), \tag{14}$$

where $s_{E_q^{(t)}}(q, c)$ is the score computed with embedding $E_q^{(t)}$ using the same inner-product function as in Eq. (7).

We refer to $k$ (the number of partitions per elimination step) as the *width* of test-time scaling, and to $d$ (the number of embedding update iterations) as its *depth*. Together, this forms a graph-based scaling procedure, where embeddings propagate along a graph of candidate partitions to refine query representations iteratively, as illustrated in Figure 1(c). In practice, both width $k$ and depth $d$ are selected based on validation performance, and the best hyperparameters are directly applied at inference time.

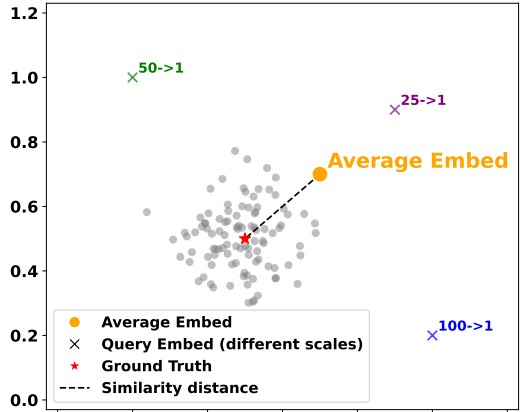

Figure 2: **The graph-based test-time ensemble produces richer query representations than a single embedding.** t-SNE visualizations show that averaged embeddings from LRanker tend to lie closer to the ground-truth item.

### 3.4 QUALITATIVE ILLUSTRATION

We provide a qualitative illustration in Figure 2 to show how LRanker 's test-time scaling mechanism leverages multiple query embeddings to enhance representation quality compared to using a single embedding in a dual-tower setup. On a recommendation dataset with 100 candidates, we apply t-SNE to visualize candidate embeddings, the ground-truth item embedding, and the query embeddings produced at different $N \to 1$ scales. The plots indicate that the averaged query embedding from test-time ensemble integrates complementary strengths from individual embeddings and qualitatively appears closer to the ground-truth, providing intuitive evidence for its potential to improve ranking performance.

## 4 EXPERIMENTS

To explore the capability of LLMs for large-candidate ranking, we conduct a comprehensive training and evaluation of the proposed LRanker across seven interdisciplinary tasks in LLM ranking bench

(RBench) with varying candidate scales. We then compare its performance against both general ranking baselines and domain-specific methods. We begin by introducing the tasks within the LLM ranking framework.

**Task description**. The details of the tasks are summarized across three scenarios in Table 2.

- **RBench-Large.** For the large-candidate ranking scenario, we employ four widely used datasets, where the number of candidates ranges from several thousand to over ten thousand as shown in Table 2. We begin our experiments with two sequential recommendation datasets, MovieLens ml-1m (Rec-Movie) (Hou et al., 2024b) and Amazon Toys (Rec-Toy) (McAuley et al., 2015; Ni et al., 2019). For both tasks, following prior studies (Geng et al., 2022; Hua et al., 2023), we construct each sample by extracting 20 consecutive interactions as the historical sequence, while designating the 21st interaction as the ground-truth item. For evaluation, we adopt the widely used leave-one-out strategy. In addition, we adopt datasets from passage ranking and product search tasks, namely MS MARCO (Bajaj et al., 2016) and ESCI (Reddy et al., 2022), respectively. For both datasets, we assign one positive sample to each query and construct the negative sample set for each query by aggregating the negative samples from all queries. We split the data into training, validation, and test sets using an 8:1:1 ratio.

- **RBench-Ultra.** This scenario is designed to investigate the capability limits of the LLM-based `LRanker` in addressing ultra-large ranking tasks with candidate pools at the million scale. To this end, we employ the sequential recommendation dataset Amazon Clothing (Rec-Clothing) (McAuley et al., 2015; Ni et al., 2019), which contains nearly 7 million candidate items as shown in Table 2. For this task, we follow the same setting as in the RBench-Large setup, namely extracting 20 consecutive interactions as the historical sequence and designating the 21st interaction as the ground-truth item. Evaluation is also conducted using the widely adopted leave-one-out strategy.

- **RBench-Small.** We design this scenario to explore the capability of `LRanker` in ranking scenarios with relatively small candidate sets, such as the re-ranking stage in recommender systems. To this end, following prior work (Feng et al., 2025; 2024), we adopt the sequential recommendation task Rec-Music and the LLM routing task Routing-Balance, both with 20 candidates as shown in Table 2, which is fully consistent with the setting in (Feng et al., 2025).

**Baselines and metrics**. We evaluate a variety of baseline methods across three scenarios. The baselines are categorized into two groups: **(a)** *General baselines* that apply across tasks, and **(b)** *Task-specific baselines* tailored to each task. For all methods, we primarily use Mean Reciprocal Rank (MRR) (Voorhees et al., 1999; Cremonesi et al., 2010) and Normalized Discounted Cumulative Gain (NDCG@K) (Järvelin & Kekäläinen, 2002; Burges et al., 2005; Liu et al., 2009) with K = 10 to evaluate ranking performance in the main text.

- **General baselines.** We consider two categories of general baselines: retrieval-based methods and LLM-based methods. In retrieval-based methods, user queries or histories are treated as the query, while candidates are regarded as the corpus. We employ both a classical probabilistic retrieval model and a modern dense retrieval model: 1) *BM25* (Robertson et al., 2009), a traditional probabilistic retrieval function that leverages term frequency, inverse document frequency, and document length normalization. 2) *Contriever* (Izacard et al., 2021), a state-of-the-art dense retrieval model trained with contrastive learning and hard negatives.

- **Task-specific baselines.** For the **recommendation tasks in scenarios of RBench-Large and RBench-Ultra**, following prior work on large-scale recommendation (Rajput et al., 2023), we implemented five sequential recommendation baselines: 1) *FM* (Rendle, 2010): A general predictive model that efficiently captures all pairwise feature interactions, widely used as a strong baseline for recommendation and CTR prediction. 2) *BERT4Rec* (Sun et al., 2019): A sequential recommender that applies the bidirectional Transformer (BERT) architecture to user–item sequences, enabling effective modeling of complex item dependencies. 3) *GRU4Rec* (Hidasi et al., 2015): A session-based recommendation model that leverages gated recurrent units (GRUs) to capture sequential dependencies in user interaction data. 4) *SASRec* (Kang & McAuley, 2018): A Transformer-based sequential recommender that employs self-attention to model both short- and long-term user preferences. 5) *Tiger* (Rajput et al., 2023): A state-of-the-art generative retrieval framework designed for large-scale recommendation, which encodes items into semantic IDs and autoregressively generates them for efficient ranking under massive candidate sets.

Table 3: **Model performance comparison with general ranking baselines and task-specific baselines across four scenarios on NDCG@10 and MRR**. Left: Rec-Movie and Rec-Toy. Right: MS MARCO and ESCI. **Bold** and underline denote the best and second-best results.

| Model | Rec-Movie | | Rec-Toy | | Model | MS MARCO | | ESCI | |
|---|---|---|---|---|---|---|---|---|---|
| | NDCG@10 | MRR | NDCG@10 | MRR | | NDCG@10 | MRR | NDCG@10 | MRR |
| *General Ranking Baselines* | | | | | *General Ranking Baselines* | | | | |
| BM25 | 0.18 | 0.54 | 0.37 | 0.42 | BM25 | 34.77 | 26.28 | 33.70 | 23.77 |
| Contriever | 0.24 | 0.43 | 0.84 | 1.11 | Contriever | 44.36 | 33.29 | 29.41 | 26.17 |
| *Task-specific Baselines* | | | | | *Task-specific Baselines* | | | | |
| FM | 2.35 | 2.01 | 0.95 | 0.98 | RankBERT-110M | 42.26 | 28.59 | 42.39 | 31.45 |
| BERT4Rec | 4.08 | 3.56 | 1.26 | 1.31 | Multilingual-E5-560M | 53.73 | 46.49 | 53.17 | 48.43 |
| GRU4Rec | 4.12 | 3.59 | 1.59 | 1.46 | KaLM-mini-instruct-0.5B | 50.57 | 40.07 | 55.21 | 50.28 |
| SASRec | 4.36 | 3.84 | 1.65 | 1.52 | BGE-Rerank-v2-m3-568M | 53.43 | 47.74 | 52.02 | 48.10 |
| Tiger | 7.37 | 6.12 | 2.99 | 2.33 | RankLLaMA 8B | 52.22 | 48.83 | 55.78 | 52.37 |
| LRanker | **8.02** | **7.80** | **3.21** | **2.42** | LRanker | **54.80** | **49.28** | **58.80** | **57.01** |

For the **recommendation tasks in RBench-Small scenario**, we follow the baseline setup in (Feng et al., 2025) and adopt three representative methods: 1) *SASRec* (Kang & McAuley, 2018): A self-attention-based sequential recommender that models users' sequential behavioral patterns using a Transformer architecture. 2) *BPR* (Rendle et al., 2012): A pairwise ranking method that optimizes sequential recommendation by encouraging observed items to be ranked higher than unobserved ones. 3) *R1-Rec* (Lin et al., 2025): A reinforcement learning-based framework that directly optimizes retrieval-augmented LLMs for recommendation tasks using downstream feedback. As for the **routing task**, we compared three representative routers: 1) *RouterKNN* (Hu et al., 2024): A simple yet effective routing baseline that assigns queries to models by retrieving similar examples and applying majority voting. 2) *RouterBERT* (Ong et al., 2024): A lightweight BERT model fine-tuned for routing decisions using classification over task labels. 3) *GraphRouter* (Feng et al., 2024): A state-of-the-art graph-based router that balances performance and cost through structural modeling.

Finally, for the **passage ranking and product search tasks**, we implemented three specialized ranking baselines: 1) *RankBERT-110M* (Nogueira & Cho, 2019): A BERT-based passage reranker fine-tuned on MS MARCO relevance judgments, treating ranking as a binary classification problem. 2) *Multilingual-E5-560M* (Wang et al., 2022): A multilingual embedding model optimized for retrieval and ranking tasks, trained with contrastive learning objectives to generate semantically meaningful embeddings across languages. 3) *KaLM-mini-instruct-0.5B* (Hu et al., 2025): A 0.5B-parameter multilingual embedding model, instruction-tuned for retrieval and ranking tasks. 4) *BGE-Rerank-v2-m3-568M* (Xiao et al., 2024): A state-of-the-art reranker from the BGE series, fine-tuned on large-scale relevance datasets to enhance cross-encoder-based ranking performance. 5) *RankLLama-8B* (Ma et al., 2024): A ranking-specialized version of Llama-2-8B fine-tuned for passage ranking using pairwise and listwise objectives.

## 4.1 LRANKER OUTPERFORMS GENERAL RANKING METHODS AND TASK-SPECIFIC BASELINES

In the RBench-Large scenario, we compare the **0.6B-sized** LRanker with both general ranking baselines and task-specific baselines across four tasks—Rec-Movie, Rec-Toy, MS MARCO, and ESCI—as shown in Table C.3. Across all four tasks, LRanker consistently outperforms the strongest existing baselines by relative margins ranging from about 3% to nearly 9% in MRR, establishing clear SoTA performance. In the recommendation setting (Rec-Movie, Rec-Toy), where specialized sequential models such as Tiger dominate, LRanker still secures 7–9% relative improvements, showing that its centroid-based design provides complementary advantages even when strong temporal signals are available. In the retrieval setting (MS MARCO, ESCI), where large-scale candidate pools pose significant efficiency and quality challenges, LRanker achieves 3–9% relative gains over the best LLM rerankers (e.g., RankLLaMA, BGE-Rerank). Notably, the larger improvement on ESCI highlights LRanker's robustness in multilingual and noisy e-commerce search scenarios. Together, these results confirm that LRanker not only scales effectively across candidate sizes but also generalizes well across both recommendation and retrieval domains, consistently surpassing both traditional IR methods and task-specific LLM-based rankers.

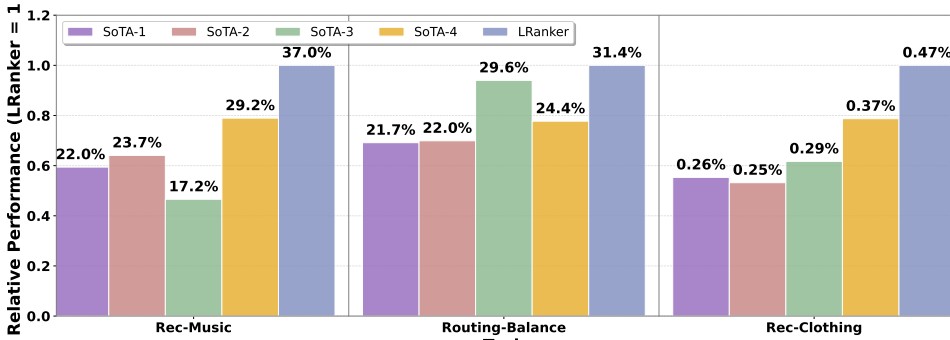

Figure 3: **Compared with state-of-the-art domain-specific baselines, `LRanker` consistently outperforms them across both ultra-long and ultra-short scenarios.** We compared the performance of `LRanker` against four representative SOTA methods across three tasks. Among them, Rec-Music and Routing-Balance are tasks in the RBench-Small scenario, while Rec-Clothing is a task in the RBench-Ultra scenario. Specifically, SOTA-1, SOTA-2, SOTA-3, and SOTA-4 correspond to SASRec (Kang & McAuley, 2018), BPR (Rendle et al., 2012), R1-Rec (Lin et al., 2025), and IRanker (Feng et al., 2025) in the Rec-Music task; GraphRouter (Feng et al., 2024), RouterBert (Ong et al., 2024), RouterKNN (Hu et al., 2024), and IRanker (Feng et al., 2025) in the Routing-Balance task; BERT4Rec (Kang & McAuley, 2018), GRU4Rec (Hidasi et al., 2015), SASRec (Nogueira et al., 2020), and Tiger (Rajput et al., 2023) in the Rec-Clothing task.

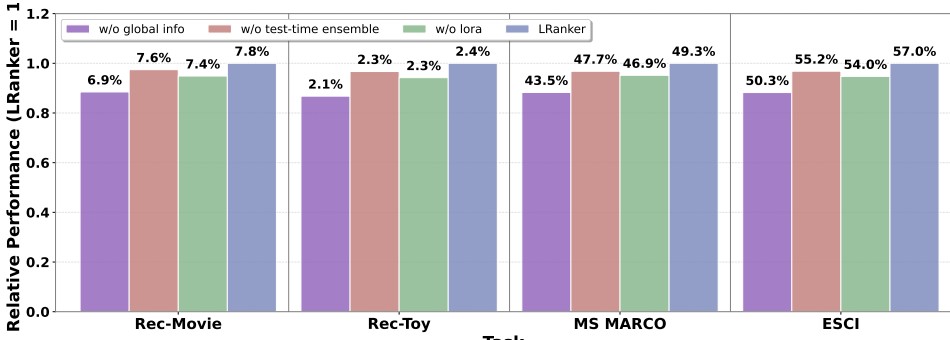

Figure 4: **Ablation studies confirm that each component of `LRanker` contributes positively to the overall performance.** To further examine their roles, we evaluate three ablated settings: (i) w/o global info removes aggregated candidate information, excluding the clustered embedding input and its projector; (ii) w/o test-time ensemble disables the ensemble mechanism, relying only on the initial embedding from the LLM; and (iii) w/o LoRA freezes LLM parameters during training and only fine-tunes the projector. As shown across Rec-Movie, Rec-Toy, MS MARCO, and ESCI, removing any component consistently leads to performance degradation.

## 4.2 `LRANKER` ACHIEVES SUPERIOR RESULTS IN BOTH RBENCH-ULTRA AND RBENCH-SMALL SCENARIOS

In the RBench-Small and RBench-Ultra scenarios, we compare the 0.6B-sized `LRanker` with representative state-of-the-art domain-specific baselines across three tasks—Rec-Music, Routing-Balance, and Rec-Clothing—as illustrated in Figure 3. `LRanker` consistently establishes superior performance over all baselines. On RBench-Small, `LRanker` achieves substantial relative improvements, with gains of up to 37% on Rec-Music and over 30% on Routing-Balance compared with the strongest sequential and routing-specific baselines. These results highlight that even in short-context ranking settings with small candidate pools, `LRanker` delivers clear benefits beyond specialized architectures such as SASRec, IRanker, and GraphRouter. On RBench-Ultra, where Rec-Clothing involves over 6.8M candidates, `LRanker` still surpasses highly optimized sequential recommenders (e.g., BERT4Rec, GRU4Rec, Tiger) by 20–30% in relative performance, underscoring its scalability to extreme candidate sizes. Overall, these findings confirm that `LRanker` not only excels in ultra-short candidate scenarios but also scales effectively to ultra-large tasks, demonstrating versatility across diverse ranking regimes.

### 4.3 ABLATION STUDIES CONFIRM THE EFFECTIVENESS OF LRANKER 'S KEY COMPONENTS

To provide a comprehensive understanding of the key components of `LRanker`, we conduct a series of experiments to investigate the effect of different components.

- **w/o global info**: Evaluates the contribution of incorporating global candidate information. This variant removes the clustered embedding input and its associated projector from the `LRanker` framework.
- **w/o test-time ensemble**: Assesses the impact of the test-time ensemble mechanism. In this setting, `LRanker` performs ranking solely using the initial embedding generated by the LLM.
- **w/o LoRA**: Examines the role of LoRA-based training. Here, the LLM parameters are frozen during training, and only the projector is fine-tuned.

We report the evaluation results on Rec-Movie, Rec-Toy, MS MARCO, and ESCI datasets in Figure 4. It can be observed that removing global candidate information causes a clear degradation across all tasks. Without clustered embeddings and the corresponding projector, the model fails to capture global context, which limits its ability to discriminate among candidates and lowers ranking accuracy. Removing the test-time ensemble mechanism also leads to reduced performance. Without the ensemble, the model relies solely on a single embedding from the LLM, which weakens its adaptability to task-specific variations and reduces robustness. Nevertheless, the results without this mechanism remain reasonably strong, suggesting that the ensemble mainly serves as a performance booster. This indicates that while test-time ensemble can further improve ranking quality, users who prioritize inference efficiency may choose to omit it with only a modest loss in performance. This highlights the flexibility of LRanker in accommodating different application needs. Eliminating LoRA-based training likewise results in a performance drop. Freezing the LLM parameters and only fine-tuning the projector prevents the model from learning task-specific adaptations, making it harder to fully exploit the LLM's representational capacity.

## 5 ADDITIONAL RELATED WORK

Recent works have explored leveraging large language models (LLMs) for ranking under different paradigms. Token-space ranking methods treat LLMs as text rankers by converting queries and candidates into textual prompts, either through iterative elimination (IRanker (Feng et al., 2025), PRP (Qin et al., 2023)) or one-shot generation (DRanker (Feng et al., 2025), RankVicuna (Pradeep et al., 2023)). However, these approaches face efficiency and context length limitations for large candidate sets. Embedding-based paradigms address this: single-tower methods (RankLLaMA (Ma et al., 2024)) use neural scoring heads but require independent candidate scoring, while dual-tower methods improve efficiency through pre-computed embeddings but limit expressiveness with single query embeddings. Generative LLM approaches have shown strong performance across diverse ranking tasks (Liu et al., 2024a; Sun et al., 2023b; Yoon et al., 2024; Chen et al., 2025a; Hou et al., 2024c). Prompting-based methods (PRP (Qin et al., 2023), LLM4Rec (Hou et al., 2024b)) leverage LLM generalization with minimal modification, while instruction tuning approaches (GPT4Rec (Li et al., 2023a), RankRAG (Yu et al., 2024)) fine-tune models for domain-specific ranking signals. These works highlight LLMs' potential as general-purpose rankers while exposing limitations in efficiency, scalability, and complex reasoning.

## 6 CONCLUSION

We propose `LRanker`, a framework designed to address the challenges of large-candidate ranking with LLMs by integrating candidate aggregation and graph-based test-time scaling. Extensive experiments across three scenarios in *RBench* demonstrate that `LRanker` consistently outperforms existing approaches, achieving substantial improvements from small-scale to million-level candidate pools. Ablation studies further validate the effectiveness of its key components. In future work, we plan to extend `LRanker` to a broader range of ranking tasks, further exploring its generality and applicability in real-world settings.

ETHICS STATEMENT

All authors of this paper have read and adhered to the ICLR Code of Ethics. Our work does not involve human subjects, personal data, or sensitive attributes. We followed best practices for data usage, ensured compliance with licensing terms, and considered potential risks of bias or misuse.

REPRODUCIBILITY STATEMENT

We have made every effort to ensure the reproducibility of our results. Details of the model architecture, training settings, and hyperparameters are described in Section 4. All datasets we used are publicly available. The training scripts and evaluation code will be released upon publication to facilitate replication.

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

## A  PROMPT USAGE

To unify the input format across different tasks, we design prompt templates that integrate the user query with the aggregated candidate information through the special token `<|embedding|>`. These templates guide the model to attend not only to the query but also to the global context of candidate representations. Specifically, we construct task-specific templates for four representative tasks: Recommendation (Table 4), Routing (Table 5), Passage Ranking (Table 6), and Product Searching (Table 7). Each template follows a unified structure but adapts the final instruction to match the objective of the corresponding task.

Table 4: **Prompt template for Recommendation task.**

| |
|---|
| Task: Recommendation |
| Query: [USER QUERY] |
| `<|embedding|>` Based on the global context information (candidate items) and the query above, recommend the most relevant item. |

Table 5: **Prompt template for Routing task.**

| |
|---|
| Task: Routing |
| Query: [USER QUERY] |
| `<|embedding|>` Based on the global context information (candidate LLMs/agents) and the query above, select the most suitable route or model. |

Table 6: **Prompt template for Passage Ranking task.**

| |
|---|
| Task: Passage Ranking |
| Query: [USER QUERY] |
| `<|embedding|>` Based on the global context information (candidate passages) and the query above, identify the most relevant passage. |

## B  IMPLEMENTATION DETAILS

We implement `LRanker` on top of the Qwen3-0.6B Embedding model[2] using LoRA adaptation. Before training, we generate 1024-dimensional offline candidate embeddings for each candidate's context via Qwen3-0.6B. For all compared baselines, to ensure a fair comparison, we use the same 1024-dimensional offline candidate embeddings as LRanker for their candidate representations. For each query, we construct 10 random splits of its associated candidate set and obtain the corresponding embeddings. During training, the split candidate embeddings are clustered using k-means implemented via scikit-learn[3], producing candidate centroid embeddings that serve as global structural features. These centroids are further projected through a Linear–BatchNorm–ReLU block to 1024 dimensions and fused with the base encoder's textual embedding to form the final query representation. Training is performed with InfoNCE loss (temperature = 0.15), contrasting positives against sampled negatives. The model is trained for 15 epochs using the AdamW optimizer ($\beta_1 = 0.9$, $\beta_2 = 0.999$, weight decay = 0.01). We use a learning rate of $1 \times 10^{-4}$ with a 10% linear warm-up followed by cosine decay, and a batch size of 20. LoRA is applied to both attention and feed-forward layers with rank = 32, $\alpha = 64$, and dropout = 0.1. To further improve efficiency and stability, we enable BF16

---

[2] https://huggingface.co/Qwen/Qwen3-0.6B
[3] https://scikit-learn.org/stable/getting_started.html

Table 7: **Prompt template for Product Searching task.**

---

Task: Product Searching

Query: [USER QUERY]

`<|embedding|>` Based on the global context information (candidate products) and the query above, return the product that best matches the search intent.

---

training, gradient checkpointing, and gradient clipping (norm = 0.5). For evaluation, we determine the best graph depth and width using the validation set, and fix these configurations when testing on the held-out test set. Note that, to ensure inference efficiency, we restrict the search depth to 0–5 and the search width to 0–10. The detailed graph despth and width settings can be seen in Table 16 of Appendix. Moreover, during training we carefully designed parallelized processing that allows multiple queries and multiple partition-embedding plans to run simultaneously, significantly reducing inference latency. All experiments are conducted on 1 NVIDIA A6000 GPUs.

To improve clustering efficiency under large-scale candidate pools, we adopt two optimizations. First, we use MiniBatchKMeans, which processes the full candidate set in small batches to accelerate convergence. Second, because Qwen3-Embedding is trained with Matryoshka Representation Learning (MRL) (Zhang et al., 2025), we can obtain most semantic information by truncating its embedding to the first 128 dimensions. We therefore compute cluster assignments using only the truncated 128-dimensional embeddings, and then compute the final centroid embeddings using the full 1024-dimensional vectors. These two strategies significantly accelerate clustering and make `LRanker` scalable in real-world large-candidate settings.

# C   GENERALIZATION EXPERIMENTS

## C.1   GENERALIZATION TO NEW DATASETS

Table 8: **Model zero-shot performance comparison with general ranking baselines and task-specific baselines on Video Games and Software**. Specifically, we evaluate the method trained on the Rec-Toy dataset in RBench in a zero-shot manner on the Video Games and Software datasets. **Bold** and underline denote the best and second-best results.

| | Video Games | | Software | |
|---|---|---|---|---|
| **Model** | NDCG@10 | MRR | NDCG@10 | MRR |
| ***General Ranking Baselines*** | | | | |
| BM25 | 0.39 | 0.36 | 0.56 | 0.51 |
| Contriever | 0.90 | 0.93 | 0.64 | 0.67 |
| ***Task-specific Baselines*** | | | | |
| FM | 1.03 | 1.09 | 3.15 | 3.77 |
| BERT4Rec | 1.18 | 1.38 | 2.97 | 2.31 |
| GRU4Rec | 1.15 | 1.27 | 4.89 | 4.31 |
| SASRec | 1.21 | 1.40 | 2.83 | 2.56 |
| Tiger | 1.93 | 2.17 | 4.58 | 3.94 |
| LRanker | **2.31** | **2.61** | **5.43** | **4.86** |

To evaluate the generalization ability of `LRanker` on datasets beyond RBench, we first train all methods on Rec-Toy and then perform zero-shot testing on the Video Games and Software datasets (McAuley et al., 2015; Ni et al., 2019) from amazon (see Table 2 for dataset details). We report the results in 8. As shown in the table, `LRanker` exhibits clear cross-domain generalization, outperforming both general-ranking and task-specific baselines by substantial margins. On Video Games, `LRanker` delivers roughly 20% improvements over the strongest task-specific baseline

and well over 100% gains compared with general-ranking baselines. On Software, the zero-shot advantage becomes even larger, with `LRanker` surpassing the best task-specific method by around 20–25% and general-ranking baselines by several-fold. These consistent percentage gains across two unseen domains demonstrate that the ranking patterns learned from Rec-Toy transfer effectively, highlighting the robust zero-shot generalization capability of `LRanker`.

## C.2 PERFORMANCE ANALYSIS IN SCENARIOS WITH EXTREMELY IRRELEVANT CANDIDATES

Table 9: **Performance comparison in scenarios with extremely irrelevant candidates across two scenarios on NDCG@10 and MRR**. Left: Rec-Toy. Right: MS MARCO. Specifically, we train the models on the training sets of Rec-Toy and MS MARCO, where the candidates used during training are the original candidates of each dataset. During testing, however, we replace the candidates with a mixed pool that combines all candidates from Rec-Movie, Rec-Toy, MS MARCO, and ESCI. In addition, $\Delta$ performance denotes the relative difference in MRR between the results obtained under the mixed-candidate setting and those obtained under the original candidate set. **Bold** and underline denote the best and second-best results.

| | **Rec-Toy** | | | **MS MARCO** | |
|---|---|---|---|---|---|
| **Model** | NDCG@10 | MRR | **Model** | NDCG@10 | MRR |
| *General Ranking Baselines* | | | *General Ranking Baselines* | | |
| BM25 | 0.02 | 0.17 | BM25 | 30.24 | 24.87 |
| Contriever | 0.12 | 0.21 | Contriever | 39.21 | 31.09 |
| *Task-specific Baselines* | | | *Task-specific Baselines* | | |
| FM | 0.37 | 0.34 | RankBERT-110M | 36.27 | 28.13 |
| BERT4Rec | 0.55 | 0.57 | Multilingual-E5-560M | 45.85 | 41.35 |
| GRU4Rec | 0.56 | 0.60 | KaLM-mini-instruct-0.5B | 43.19 | 38.36 |
| SASRec | 0.78 | 0.64 | BGE-Rerank-v2-m3-568M | 45.83 | 43.98 |
| Tiger | 2.25 | 1.91 | RankLLaMA 8B | 46.91 | 44.04 |
| LRanker | **2.43** | **2.06** | LRanker | **49.03** | **46.40** |
| $\Delta$ *performance* | | | $\Delta$ *performance* | | |
| $\Delta$ Tiger | −24.6% | −18.0% | $\Delta$ RankLLaMA 8B | −10.2% | −10.5% |
| $\Delta$ LRanker | −24.2% | −14.9% | $\Delta$ LRanker | −9.8% | −5.9% |

To evaluate the performance of `LRanker` in scenarios with extremely irrelevant candidates, we construct a mixed candidate pool by combining all candidates from Rec-Movie, Rec-Toy, MS MARCO, and ESCI. We then compare all methods trained on the original candidate sets of Rec-Toy and MS MARCO but tested on the mixed candidate pool. The results are shown in Table 9. We can observe that although all models experience performance degradation when exposed to a large number of irrelevant candidates, `LRanker` remains consistently the most robust across both scenarios. On Rec-Toy, the drop of `LRanker` is much smaller than that of the strongest task-specific baseline, while still retaining a clear performance advantage. On MS MARCO, the relative degradation of `LRanker` is substantially lower than that of the strongest baseline, indicating that the ranking patterns learned during training generalize more effectively under heavy distribution shift. These trends demonstrate that `LRanker` not only achieves the best overall performance but also maintains superior stability and robustness in the presence of large-scale irrelevant candidates.

## C.3 EXPERIMENT UNDER CANDIDATES DISTRIBUTION SHIFT

To evaluate the robustness of `LRanker` under candidate distribution shift, we construct two separate mixed candidate pools: one combining candidates from Rec-Movie and Rec-Toy, and the other combining candidates from MS MARCO and ESCI. We then compare all methods that are trained on the original candidate sets of Rec-Toy and MS MARCO but tested on their corresponding mixed candidate pools. The results are shown in Table 10. We can observe that most baselines experience substantial performance degradation when evaluated on the mixed candidate pools, indicating their limited robustness to candidate distribution shift. In contrast, `LRanker` consistently achieves the highest accuracy under both Rec-Toy and MS MARCO settings and exhibits a significantly smaller performance drop compared to strong task-specific baselines such as Tiger and RankLLaMA 8B. These results demonstrate that `LRanker` effectively leverages global candidate information and maintains stable ranking behavior even when the candidate distribution changes at test time.

Table 10: **Performance comparison in scenarios under candidates distribution shift across two scenarios on NDCG@10 and MRR**. Left: Rec-Toy. Right: MS MARCO. Specifically, we train the models on the training sets of Rec-Toy and MS MARCO, where the candidates used during training are the original candidates of each dataset. During testing, we replace the candidate set of Rec-Toy with a mixed candidate pool constructed from both Rec-Movie and Rec-Toy. Similarly, for MS MARCO and ESCI, we replace each candidate set with mixed candidate pools that combine candidates from MS MARCO and ESCI. In addition, $\Delta$ performance denotes the relative difference in MRR between the results obtained under the mixed-candidate setting and those obtained under the original candidate set.**Bold** and underline denote the best and second-best results.

| | **Rec-Toy** | | | **MS MARCO** | |
| --- | --- | --- | --- | --- | --- |
| **Model** | NDCG@10 | MRR | **Model** | NDCG@10 | MRR |
| *General Ranking Baselines* | | | *General Ranking Baselines* | | |
| BM25 | 0.14 | 0.29 | BM25 | 32.61 | 27.22 |
| Contriever | 0.54 | 0.52 | Contriever | 40.68 | 33.74 |
| *Task-specific Baselines* | | | *Task-specific Baselines* | | |
| FM | 0.71 | 0.48 | RankBERT-110M | 38.52 | 29.59 |
| BERT4Rec | 1.22 | 1.28 | Multilingual-E5-560M | 47.95 | 45.21 |
| GRU4Rec | 1.26 | 1.43 | KaLM-mini-instruct-0.5B | 45.73 | 39.48 |
| SASRec | 1.39 | 1.34 | BGE-Rerank-v2-m3-568M | 47.06 | 46.59 |
| Tiger | 2.34 | 2.27 | RankLLaMA 8B | 47.08 | 47.27 |
| LRanker | **2.54** | **2.36** | LRanker | **51.41** | **49.01** |
| $\Delta$ *performance* | | | $\Delta$ *performance* | | |
| $\Delta$ Tiger | $-30.2\%$ | $-6.2\%$ | $\Delta$ RankLLaMA 8B | $-9.8\%$ | $-3.2\%$ |
| $\Delta$ LRanker | $-20.9\%$ | $-2.5\%$ | $\Delta$ LRanker | $-6.2\%$ | $-0.55\%$ |

# D  EXPERIMENTS ON SCALABILITY

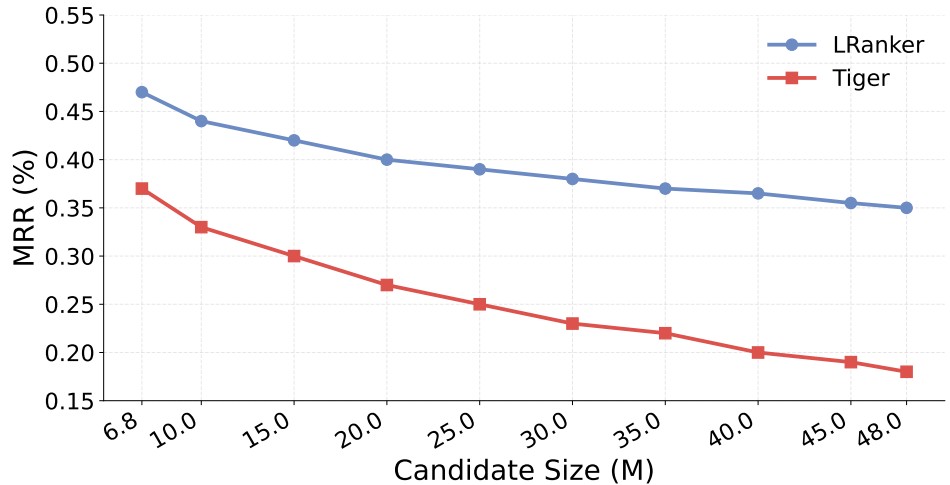

Figure 5: **The change in MRR performance of LRanker and Tiger as the candidate size increases.** Specifically, we use increments of 5M candidates and scale up to a maximum of 48M candidates.

To examine the limits of LRanker in handling extremely large candidate sets and to analyze how its performance changes under such conditions, we conduct experiments on the Amazon-23 dataset (Hou et al., 2024a), which contains approximately 4.8M candidates. Specifically, we take LRanker and Tiger trained on Rec-Clothing from RBench-Ultra and progressively expand the candidate pool by randomly adding candidates in increments of 5M on top of the original pool, evaluating the MRR performance at each step. We report the results in Figure 5. We can observe that both LRanker and Tiger exhibit consistent performance degradation as the candidate size increases, but LRanker

maintains a noticeably slower relative decay. From 6.8M to 48M candidates, LRanker's MRR decreases by roughly **25%** relative to its initial value, whereas Tiger suffers a substantially larger relative drop of around **50%**. Moreover, the degradation curves of both methods follow the classic *IR scaling after saturation* behavior: once the candidate pool grows beyond a certain scale, the decline in ranking performance becomes progressively flatter rather than continuing linearly. This saturation effect likely occurs because, as the candidate pool grows, most newly added items are increasingly irrelevant to the query and therefore less confusable with the ground-truth item. In high-dimensional embedding spaces, the number of true hard negatives grows sublinearly with corpus size, while the proportion of far, irrelevant items dominates. As a result, performance degradation slows and eventually plateaus.

# E ADDITIONAL ABLATION STUDIES

## E.1 COMPARATIVE STUDY OF K-MEANS AND ALTERNATIVE CLUSTERING TECHNIQUES

Table 11: **Performance comparison of the possible candidate aggregation encoder variants across four tasks**.

| Model | Rec-Movie MRR | Rec-Toy MRR | Model | MS MARCO MRR | ESCI MRR |
|---|---|---|---|---|---|
| Set Encoder | 6.20 | 1.95 | Set Encoder | 43.50 | 50.10 |
| PCA | 6.70 | 2.05 | PCA | 45.50 | 52.40 |
| Hierarchical Clustering | 7.00 | 2.18 | Hierarchical Clustering | 46.80 | 53.80 |
| LRanker | **7.80** | **2.42** | LRanker | **49.28** | **57.01** |

In this section, we compare LRanker with other methods based on different candidate aggregation methods. To be specific, we design three baselines. To ensure a fair comparison between LRanker and the baselines, we constrain all methods such that the final candidate embeddings fed into the LLM occupy the same number of "tokens".

- **Set Encoder**: In this setting, we sample $K$ candidates from the full candidate pool and pass their embeddings through a cross-attention module. The resulting representations are then fed into the LLM. Here, $K$ is set to match the number of k-means cluster centroids used in LRanker.
- **PCA**: In this setting, the offline embeddings of candidates are first reduced to 256 dimensions using PCA, followed by k-means clustering.
- **Hierarchical Clustering**: Compared with LRanker, in this setting, we replace k-means with hierarchical clustering while keeping the number of clusters unchanged.

As shown in Table 11, LRanker consistently surpasses all three aggregation baselines across all tasks. The Set Encoder performs the worst because it samples only a small subset of candidates and aggregates them through cross attention, inevitably discarding global information from the full candidate pool. The PCA baseline performs better than Set Encoder but still suffers from substantial information loss due to projecting 1024-dimensional embeddings into a 256-dimensional space prior to clustering. Hierarchical Clustering achieves the strongest baseline performance and comes closest to LRanker, as it preserves more structural relationships and avoids sampling or dimensionality reduction. However, its computational cost is prohibitive: agglomerative hierarchical clustering requires $O(n^2 d)$ time and $O(n^2)$ memory to compute and store all pairwise distances, where $n$ is the number of candidates and $d$ is the embedding dimensionality, making it infeasible when $n$ reaches millions. In contrast, k-means used in LRanker scales as $O(nkdT)$, where $k$ is the number of clusters and $T$ is the number of iterations, thus providing linear rather than quadratic scaling in $n$. As described in Appendix B, LRanker further improves the efficiency of k-means by using MiniBatchKMeans, which reduces the effective complexity to $O(bkdT)$ with a small batch size $b \ll n$, and by exploiting the Matryoshka Representation Learning property of Qwen3-Embedding, which enables clustering on truncated embeddings of dimension $d' \ll d$ (e.g., 256 instead of 1024) without sacrificing semantic fidelity. These design choices allow LRanker to maintain strong ranking performance while enabling fast inference under extremely large candidate pools, making it practical for real-world deployment.

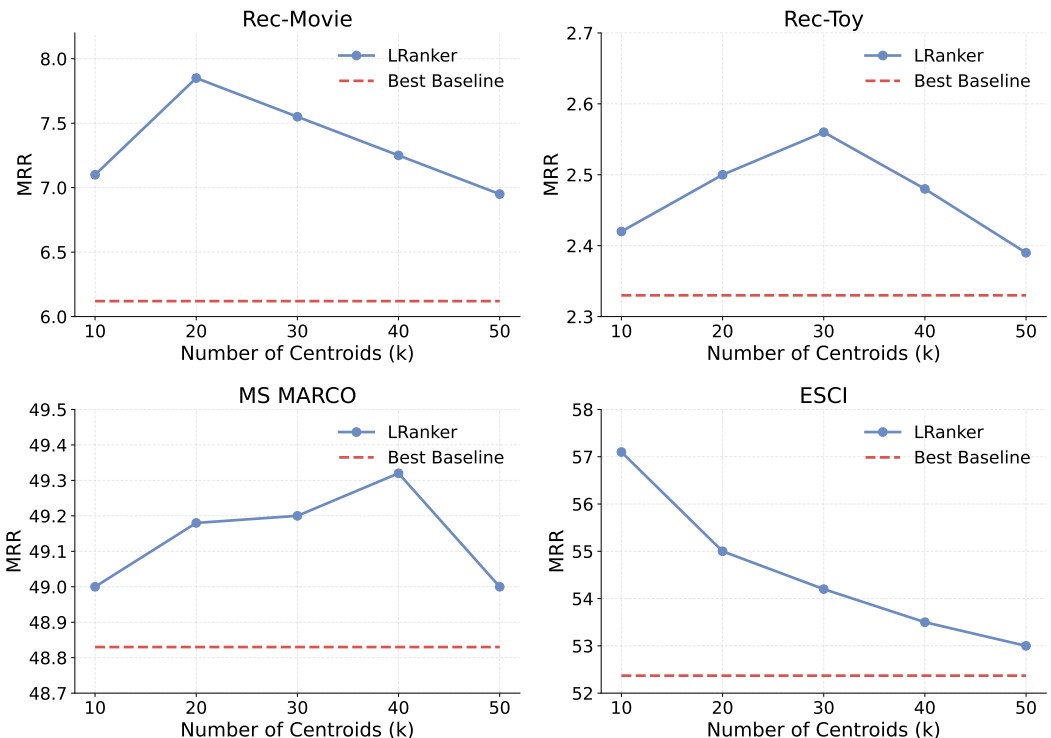

Figure 6: **Effect of the number of centroids ($k$) on the performance of `LRanker` across four tasks.** `LRanker` consistently outperforms the strongest baseline under all choices of $k$, and typically reaches peak performance at moderate values ($k = 10$–$50$). Larger $k$ introduces finer but noisier partitions, resulting in a slight performance drop.

### E.2 IMPACT OF THE CHOICE OF K ON PERFORMANCE

In this experiment, we study how the number of centroids $k$ influences the effectiveness of `LRanker` in Figure 6. From a computational perspective, $k$ directly affects both the clustering cost during preprocessing and the inference-time cost of encoding candidate-cluster features. To ensure a practical efficiency–effectiveness trade-off, we explore a moderate range of $k \in \{10, 20, 30, 40, 50\}$, which covers the values that are computationally feasible while still allowing sufficient granularity for capturing the structure of the candidate pool. Across all four tasks (Rec-Movie, Rec-Toy, MS MARCO, and ESCI), `LRanker` consistently outperforms the strongest baseline under all choices of $k$, demonstrating that the model is highly robust to the selection of this hyperparameter. Performance typically peaks at moderate values (e.g., $k = 10$–$50$), where the centroids provide a balanced level of abstraction: too few centroids underrepresent the candidate distribution, whereas excessively large $k$ yields finer but noisier partitions, leading to slight performance drops. Nevertheless, the margin over the best baseline remains substantial for all settings, illustrating that `LRanker` maintains strong effectiveness even when $k$ varies within a wide operational range.

### E.3 EFFECT OF CENTROID DIMENSIONALITY ON MODEL PERFORMANCE

We further investigate how the dimensionality of the centroid embeddings affects the performance of `LRanker`. This hyperparameter directly influences the expressiveness of the aggregated candidate representations as well as the computational cost of the clustering stage and the subsequent LLM encoding step. To balance semantic fidelity and efficiency, we evaluate centroid dimensionalities in the range $\{256, 512, 768, 1024\}$, which spans from aggressively truncated representations to the full-dimensional Qwen3-Embedding output (the principle and rationale for truncation can be found in appendix B). As shown in Figure 7, across all four tasks, `LRanker` consistently outperforms the strongest baseline for every dimensionality setting, demonstrating strong robustness to this design

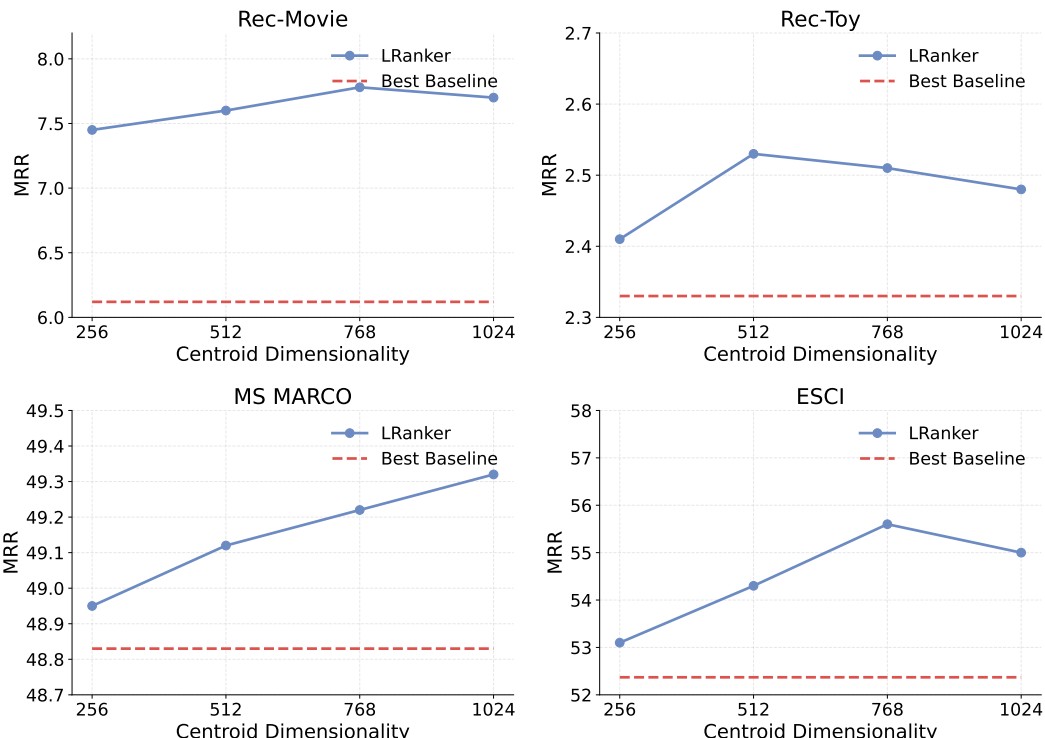

Figure 7: **Effect of the centroid dimensionality on the performance of `LRanker` across four tasks.** Increasing the dimensionality generally improves the quality of centroid representations by preserving more semantic information, leading to consistent gains over the strongest baseline under all settings. Moderate dimensions (256–1024) already achieve strong results, indicating that `LRanker` does not require the full 1024-dimensional embeddings to maintain high effectiveness.

choice. Increasing the centroid dimensionality generally leads to better performance, as higher-dimensional centroids preserve more semantic information from the original candidate embeddings. However, we observe that moderate dimensions (e.g., 512–768) already capture most of the useful structure and yield competitive or near-peak performance. This suggests that `LRanker` does not heavily rely on full 1024-dimensional centroids to achieve high ranking effectiveness. The ability to operate effectively with reduced centroid dimensionality highlights the efficiency advantages of `LRanker`, as lower-dimensional centroids reduce both clustering and inference-time computation while maintaining strong accuracy.

### E.4    ANALYSIS OF BACKBONE LLM INFLUENCE

In addition to comparing against general and task-specific baselines, we further evaluate the ranking capability of the backbone models used in `LRanker` by directly applying two Qwen3 variants (0.6B and 4B) to the Rec-Movie and Rec-Toy tasks in a zero-shot manner. As shown in Table 12, the 0.6B model performs substantially worse than most task-specific baselines, indicating that a small LLM backbone lacks sufficient inductive bias for effective item ranking. The larger 4B model shows noticeable improvement, yet still lags behind the strongest baselines (e.g., Tiger), demonstrating that simply scaling the backbone model size does not close the gap. These results highlight that directly using a pretrained Qwen3 model to solve ranking tasks does not inherently provide performance gains. In contrast, `LRanker` achieves substantial improvements by combining a carefully designed training objective with graph-based test-time scaling, which equips the LLM with candidate aggregation encoder and enables robust ranking behavior far beyond what the backbone alone can offer.

Table 12: **Evaluating the direct task-solving capability of Qwen3 backbones of different sizes against general and task-specific baselines on Rec-Movie and Rec-Toy. Bold** and underline denote the best and second-best results.

| Model | Rec-Movie | | Rec-Toy | |
|---|---|---|---|---|
| | NDCG@10 | MRR | NDCG@10 | MRR |
| *General Ranking Baselines* | | | | |
| BM25 | 0.18 | 0.54 | 0.37 | 0.42 |
| Contriever | 0.24 | 0.43 | 0.84 | 1.11 |
| *Backbone Models* | | | | |
| Qwen3 0.6B | 0.61 | 0.84 | 1.56 | 1.52 |
| Qwen3 4B | 0.84 | 1.29 | 2.52 | 2.11 |
| *Task-specific Baselines* | | | | |
| FM | 2.35 | 2.01 | 0.95 | 0.98 |
| BERT4Rec | 4.08 | 3.56 | 1.26 | 1.31 |
| GRU4Rec | 4.12 | 3.59 | 1.59 | 1.46 |
| SASRec | 4.36 | 3.84 | 1.65 | 1.52 |
| Tiger | 7.37 | 6.12 | 2.99 | 2.33 |
| LRanker | **8.02** | **7.80** | **3.21** | **2.42** |

## F COMPUTATION AND RUNTIME ANALYSIS

### F.1 EVALUATION OF COMPUTATIONAL COST AND RUNTIME AGAINST BEST BASELINES

Table 13: Computation time and memory usage for different models on Rec-Movie and MS MARCO.

| Scenario | Model | Train Time | Train Memory | Test Time | Test Memory |
|---|---|---|---|---|---|
| Rec-Movie | Tiger | 21 h 36 m | 10.8 GB | 20 min | 200 MB |
| | LRanker | 52 min | 24.5 GB | 15 min | 17.2 GB |
| MS MARCO | RankLLaMA 8B | 6 h 33 min | 188 GB | 21.67 min | 71.28 GB |
| | LRanker | 21 min | 25.0 GB | 10 min | 17.5 GB |

As shown in Table 13, we compare the computation time and memory usage of LRanker against the strongest task-specific baselines on Rec-Movie and MS MARCO under identical hardware settings. On Rec-Movie, LRanker completes training in only 52 minutes, representing more than a *24× reduction* in training time compared with Tiger (21 h 36 m), while also achieving lower test-time latency (15 min vs. 20 min). A similar trend appears on MS MARCO: LRanker requires just 21 minutes to train, in stark contrast to RankLLaMA 8B, which takes 6 h 33 m. Test-time latency is also reduced by more than half (10 min vs. 21.67 min). Although LRanker uses moderately more memory during training due to LoRA adaptation and centroid aggregation, its test-time memory footprint (17–18 GB) remains lightweight, especially compared with RankLLaMA 8B, which consumes over 70 GB. These results highlight that LRanker achieves substantial improvements in computational efficiency and latency without sacrificing effectiveness. Overall, LRanker provides a practical and scalable solution for real-world retrieval and ranking systems, where fast training and low-latency inference are essential.

### F.2 SCENARIO-WISE MEMORY USAGE OF LRanker

Table 14 reports the scenario-wise memory usage of LRanker during both training and inference across all tasks in RBench. Overall, the memory consumption remains highly stable for most scenarios. Training typically requires around 24–25 GB of GPU memory, while inference remains within 16–18 GB. This stability stems from the design of LRanker, whose memory footprint is dominated by the LoRA-adapted backbone model and the centroid aggregation module.

### F.3 LATENCY COMPARISON IN TABLE 1

Table 15 shows that LRanker delivers consistently low per-query latency across both small and large candidate pools. On Rec-Music (20 candidates), models such as PRP, IRanker, and RankGPT

Table 14: Memory Requirements of RBench Scenarios.

| Scenario | Train Memory | Inference Memory |
|---|---|---|
| Rec-Music | 24.4 GB | 16.9 GB |
| Routing-Balance | 24.4 GB | 16.9 GB |
| Rec-Movie | 24.5 GB | 17.2 GB |
| ESCI | 24.5 GB | 17.2 GB |
| Rec-Toy | 25.0 GB | 17.5 GB |
| MS MARCO | 25.0 GB | 17.5 GB |
| Rec-Clothing | 37.2 GB | 29.2 GB |

Table 15: **Per-query inference latency comparison across different models on Rec-Music and MS MARCO.**

| Scenario | Model | Per-query Inference Time |
|---|---|---|
| Rec-Music (20 candidates) | PRP | 38.3 s |
| | IRanker | 6.5 s |
| | RankGPT | 3.8 s |
| | LRanker | 15 ms |
| MS MARCO (24,697 candidates) | RankLLaMA 8B | 21.67 min |
| | LRanker | 20 ms |

require seconds of computation, while `LRanker` responds in only 15 ms. The gap widens on MS MARCO, where RankLLaMA 8B needs over 21 minutes per query due to full-candidate scoring, whereas `LRanker` maintains a 20 ms latency by operating on precomputed centroids instead of all candidates. These results demonstrate that `LRanker` is not only accurate but also two to three orders of magnitude faster than existing LLM-based rankers, making it suitable for real-time production systems.

## F.4 Impact of Widths and Depths on the Computational Efficiency of LRanker

Table 16: **Optimal width/depth settings of LRanker and their relative latency overhead compared to direct ranking.**

| Scenario | Best Width | Best Depth | Latency Increase vs. Direct Ranking (%) |
|---|---|---|---|
| **RBench-Small** | | | |
| Rec-Music | 3 | 3 | 1.2% |
| Routing-Balance | 3 | 3 | 1.5% |
| **RBench-Large** | | | |
| Rec-Movie | 5 | 4 | 3.2% |
| Rec-Toy | 5 | 6 | 3.8% |
| MS MARCO | 9 | 6 | 6.5% |
| ESCI | 6 | 5 | 5.2% |
| **RBench-Ultra** | | | |
| Rec-Clothing | 10 | 6 | 7.8% |

As shown in Table 16, the additional latency introduced by the graph-based test-time scaling module is surprisingly small across all scenarios. Even when the best-performing configurations require moderate widths and depths (e.g., width = 9, depth = 6 on MS MARCO), the relative latency increase over direct ranking remains below 8%, and is often as low as 1–3% on the smaller RBench tasks. This demonstrates that LRanker achieves substantial ranking improvements with minimal overhead. Moreover, when viewed in absolute terms, LRanker remains extremely fast. As reported in Appendix F.3, its per-query inference latency is already orders of magnitude lower than that of strong LLM-based rankers (e.g., RankLLaMA 8B), and remains competitive even against lightweight models

such as RankGPT. This high efficiency is enabled by the test-time design described in Appendix B: MiniBatchKMeans for scalable centroid construction, Matryoshka Representation Learning (MRL) for low-dimensional clustering, and centroid-only aggregation for compact LLM input. Together, these optimizations ensure that LRanker maintains both strong performance and low latency, even when operating under larger width/depth configurations.

## G LLM WRITING USAGE DISCLOSURE

An LLM was applied as a writing aid to enhance the clarity and linguistic quality of this paper, specifically by correcting grammatical errors and polishing sentence flow. No part of the research design, data analysis, or interpretation relied on the use of the LLM.

