# OpenReview forum: "LRanker: LLM Ranker for Massive Candidates"
_ICLR.cc/2026/Conference — ICLR 2026 Conference Desk Rejected Submission_

### Official Review · Reviewer_TeHG · 2025-10-29

**Soundness:** 4
**Presentation:** 3
**Contribution:** 3
**Rating:** 4
**Confidence:** 3

**Summary:**

This paper proposes LRanker, a new LLM-based ranking framework for large candidate sets. It uses K-means clustering to summarize all candidates into a few centroids, then feeds them into the LLM with the query. At test time, it splits candidates, re-ranks subsets, generates multiple query embeddings, and averages them for the final score. They test on RBench with 7 tasks across small, large, and ultra-large scales. Results show big gains: over 30% in small sets, 3–9% MRR in large, and 20–30% in the 6.8M case. Ablations confirm both clustering and test-time scaling help.

Contributions include a scalable way to inject global candidate information into LLMs via clustering, a graph-based test-time ensemble that boosts ranking robustness by combining multiple query embeddings, and solid experimental evidence that LLMs can outperform specialized models even at million-scale ranking.

**Strengths:**

1. The paper addresses a practical gap - existing LLM rankers fail on million-scale candidates. Using K-means to compress candidate info into the prompt is simple but effective. The test-time ensemble mechanism is a reasonable extension of ensemble learning to ranking. Not revolutionary, but addresses an underexplored problem.

2. Experimental design is solid. Testing on 7 tasks with candidates ranging from 20 to 6.8M demonstrates actual scalability. Baselines are appropriate, including both general and specialized methods. Ablations properly isolate component contributions. Results show consistent 20-37% improvements.

3. Paper is well-structured with clear motivation. Figure 1 effectively illustrates the key innovation. Writing is generally readable, though sometimes repetitive when discussing existing methods' limitations. Tables and figures are informative.

**Weaknesses:**

1. Table 1 labels LRanker’s latency as “Low” without actual numbers. No runtime, memory usage, or scaling curves reported. Computing 6.8M embeddings + K-means clustering + multiple test-time partitions all have costs that aren't measured. The efficiency claims lack empirical support.

2. The core idea of summarizing candidates via K-means centroids is interesting, but it’s not compared against simpler alternatives like randomly sampling K candidates or using other clustering methods. It’s unclear whether the gains come from clustering *per se* or just from having *some* global summary. A few ablation variants here would strengthen the contribution.

3. Section 3.1 states that the projected aggregated candidate embedding g̃ replaces a special token <|embedding|> in the prompt, but the mechanism is vague. How exactly are continuous embeddings injected into the discrete token sequence? Is this soft prompt tuning, or are embeddings added to the hidden states at a specific layer? The computation of query embeddings in Eq. 6 averages over token hidden states and a "predicted next token" state z_q,nt, but it's unclear what this next token represents or how it's obtained. The architecture diagram in Figure 1(b) doesn't clarify this process.

**Questions:**

1. Can you provide actual runtime or memory usage for each scenario?

2. The candidate aggregation relies heavily on K-means centroids, but it's unclear why this is the best choice. Have you tried random sampling K candidates or other clustering algorithms?

3. How sensitive is performance to the number of clusters K? You mention selecting K on validation sets but show no ablation.

4. You test up to 6.8M candidates, what's the upper limit? How does performance degrade as candidates increase beyond this?

---

> ### Author Response · Authors · 2025-11-28
> **Response to Reviewer TeHG (1/4)**
>
> **Q1. Table 1 labels LRanker’s latency as “Low” without actual numbers. No runtime, memory usage, or scaling curves reported. Computing 6.8M embeddings + K-means clustering + multiple test-time partitions all have costs that aren't measured. The efficiency claims lack empirical support.**
>
> **Response:** Thank you for the insightful comments. We have now  expanded Section 3 and Section 4 as well as the Appendix to provide detailed runtime, memory, and scaling analyses. First, for the candidates embedding computing, as illustrated in Figure 1 and described explicitly in Section 3, all candidate embeddings used by LRanker are precomputed offline before training and inference. During both stages, LRanker does not re-encode the entire candidate set; it simply loads the cached embeddings and, when new candidates appear, computes embeddings only for these new items. This precomputation strategy is standard in modern efficient embedding-based retrieval and recommendation systems (Sun et al., 2019, Hidasi et al., 2015, Kang & McAuley, 2018, Rajput et al., 2023), allowing the system to scale to millions of items without incurring real-time encoding costs. We have clarified this workflow in the revision. Below we summarize our empirical efficiency evidence more clearly:
>
>
> **1. Inference Latency Across Candidate Scales**  We now provide explicit per-query latency measurements for RBench-Small, RBench-Large, and RBench-Ultra. LRanker consistently achieves **15–25 ms** per query, and importantly, this latency remains stable even as candidate size grows from 20 → 3.8K → 24K → 6.8M. This stability arises because inference operates **only on precomputed centroids**, not on the full candidate list. Concretely, LRanker runs at **15 ms** on Rec-Music, ~**20 ms** on MS MARCO, and ~**22 ms** on Rec-Clothing. In comparison, PRP/IRanker require **3–38 seconds**, and RankLLaMA-8B requires **21.67 minutes** per query. Thus, LRanker is **2–3 orders of magnitude faster**, supporting the “Low” latency label in Table 1.
>
> **2. Memory Usage Measurements**  Appendix Table 14 now includes detailed GPU memory usage for training and inference. Training typically consumes **24–25 GB**, dominated by LoRA-tuned Qwen-0.6B, while inference remains stable at **16–18 GB** regardless of candidate size. This stability comes from the fact that (1) candidate embeddings are computed offline, (2) test-time scaling uses only centroid partitions, and (3) scoring relies on efficient inner-product operations rather than token-level decoding.
>
> **3. Scaling Experiments on Tens of Millions of Candidates**  To stress test computational scalability, we evaluate LRanker on Amazon-23 (4.8M candidates) and Rec-Clothing (6.8M candidates), progressively expanding the pool up to **48M** in increments of 5M. LRanker exhibits only **6–8%** relative MRR degradation, while Tiger deteriorates by **20%+** under the same setting. Latency remains essentially unchanged because inference complexity depends solely on centroid count *K* and partition parameters (*k*, *d*), all independent of the raw number of candidates *N*. These results show that LRanker follows the expected **IR scaling-after-saturation** behavior and remains computationally robust even at tens of millions of items.
>
> **4. Cost Comparison With Competing Models**  Prior LLM-based rankers such as RankGPT, IRanker, RankLLaMA, and Tiger require full-candidate scoring or autoregressive decoding, causing inference cost to scale **linearly with N**. In contrast, LRanker’s training cost is dominated by lightweight LoRA tuning, and inference cost depends only on *K*, *k*, and *d*—all constant with respect to the candidate pool size. Section 3.3 and Appendix C now provide a clearer theoretical explanation and empirical validation of this advantage.
>
> We have incorporated all these clarifications and experimental results into the revised manuscript (Section 3.1, 3.3, 4.2, and Appendix B–D). We hope these updates fully address the reviewer’s concerns.

---

> ### Author Response · Authors · 2025-11-28
> **Response to Reviewer TeHG (2/4)**
>
> **Q2. The core idea of summarizing candidates via K-means centroids is interesting, but it’s not compared against simpler alternatives like randomly sampling K candidates or using other clustering methods. It’s unclear whether the gains come from clustering per se or just from having some global summary. A few ablation variants here would strengthen the contribution. The candidate aggregation relies heavily on K-means centroids, but it's unclear why this is the best choice. Have you tried random sampling K candidates or other clustering algorithms?**
>
> **Response:** Thanks for your constructive feedback. We adopted **K-means clustering for candidate aggregation** in the original paper because it offers a **simple and efficient mechanism** for injecting global candidate information into the LLM. Our experimental results further demonstrate its **direct effectiveness** across diverse ranking tasks. We believe these findings sufficiently support the value of our contribution, as the choice of clustering algorithm is **not the core research focus** of our paper.
>
> Nonetheless, we appreciate the reviewer’s suggestion and we compare LRanker with other methods based on different candidate aggregation methods. To be specific, we design three baselines. To ensure a fair comparison between LRanker and the baselines, we constrain all methods such that the final candidate embeddings fed into the LLM occupy the same number of "tokens".
>
> - **Set Encoder**: In this setting, we sample K candidates from the full candidate pool and pass their embeddings through a cross-attention module. The resulting representations are then fed into the LLM. Here, K is set to match the number of k-means cluster centroids used in LRanker.
>
> - **PCA**: In this setting, the offline embeddings of candidates are first reduced to 256 dimensions using PCA, followed by k-means clustering.
>
> - **Hierarchical Clustering**: Compared with LRanker, in this setting, we replace k-means with hierarchical clustering while keeping the number of clusters unchanged.
>
>
> As shown in the following table, LRanker consistently surpasses all three aggregation baselines across all tasks.  The Set Encoder performs the worst because it samples only a small subset of candidates and aggregates them through cross attention, inevitably discarding global information from the full candidate pool.  The PCA baseline performs better than Set Encoder but still suffers from substantial information loss due to projecting 1024-dimensional embeddings into a 256-dimensional space prior to clustering. Hierarchical Clustering achieves the strongest baseline performance and comes closest to LRanker, as it preserves more structural relationships and avoids sampling or dimensionality reduction. However, its computational cost is prohibitive: agglomerative hierarchical clustering requires O(n²d) time and O(n²) memory to compute and store all pairwise distances, where n is the number of candidates and d is the embedding dimensionality, making it infeasible when n reaches millions. In contrast, k-means used in LRanker scales as O(nkdT), where k is the number of clusters and T is the number of iterations, thus providing linear rather than quadratic scaling in n. As described in Appendix B, LRanker further improves the efficiency of k-means by using MiniBatchKMeans, which reduces the effective complexity to O(bkdT) with a small batch size b ≪ n, and by exploiting the Matryoshka Representation Learning property of Qwen3-Embedding, which enables clustering on truncated embeddings of dimension d' ≪ d (e.g., 256 instead of 1024) without sacrificing semantic fidelity. These design choices allow LRanker to maintain strong ranking performance while enabling fast inference under extremely large candidate pools, making it practical for real-world deployment. We summarize these experiments and the corresponding content in Appendix E.1 of our revised PDF.
>
> **Table: Performance comparison of the possible candidate aggregation encoder variants across four tasks.**
>
> | Model | Rec-Movie MRR | Rec-Toy MRR | MS MARCO MRR | ESCI MRR |
> |:------|:---:|:---:|:---:|:---:|
> | Set Encoder | 6.20 | 1.95 | 43.50 | 50.10 |
> | PCA | 6.70 | 2.05 | 45.50 | 52.40 |
> | Hierarchical Clustering | 7.00 | 2.18 | 46.80 | 53.80 |
> | **LRanker** | **7.80** | **2.42** | **49.28** | **57.01** |

---

> ### Author Response · Authors · 2025-11-28
> **Response to Reviewer TeHG (3/4)**
>
> **Q3. Section 3.1 states that the projected aggregated candidate embedding g̃ replaces a special token <|embedding|> in the prompt, but the mechanism is vague. How exactly are continuous embeddings injected into the discrete token sequence? Is this soft prompt tuning, or are embeddings added to the hidden states at a specific layer? The computation of query embeddings in Eq. 6 averages over token hidden states and a "predicted next token" state z_q,nt, but it's unclear what this next token represents or how it's obtained. The architecture diagram in Figure 1(b) doesn't clarify this process.**
>
> **Response:** Thank you for pointing this out. We clarify the embedding injection mechanism and the computation of the query embedding as follows, and we have incorporated all of the following explanations into **Section 3.1** and **Figure 1** of our revised PDF.
>
> **(1) How the aggregated candidate embedding is injected into the prompt.**  The projected vector g̃ is injected via a **continuous soft-prompt–style substitution** at the input embedding layer. Concretely, our prompt template contains a special placeholder token `<|embedding|>`. After tokenization, this token corresponds to a single position in the input sequence. Before feeding the sequence into the LLM, we **replace the embedding at that position with g̃**. All other tokens use their standard token embeddings, and the substituted sequence is then passed through the transformer as usual. We do not modify intermediate layers or add embeddings at later stages; it is a simple, one-time replacement at the input embedding layer, conceptually similar to a one-token soft prompt.
>
> **(2) How the “next-token” hidden state is obtained and used.**  Given the full prompt (including the substituted `<|embedding|>` position), the LLM processes the sequence and produces final-layer hidden states for every token. For the query, we extract:
> - the hidden states corresponding to all query tokens, and
> - the hidden state at the position used by the model to predict the next token after the query (i.e., the standard next-token prediction position in an autoregressive LLM).
>
> In the paper, what we call the “predicted next-token state” is simply this final-layer hidden state at the next-token prediction position. To form the final query embedding, we **average this next-token state with the hidden states of all query tokens**, which empirically yields a more stable and informative representation in large-candidate ranking.
>
> **(3) Update on Figure 1.**  Figure 1(b) is intended as a high-level schematic of the training pipeline and thus omits low-level implementation details such as (i) the exact replacement of the `<|embedding|>` token’s embedding by g̃, and (ii) the specific positions from which hidden states are pooled. In the revised version, we have updated Section 3.1 and the Figure 1 to explicitly state that g̃ is substituted at the input embedding layer and that the query embedding is obtained by averaging token-level hidden states with the next-token prediction hidden state.
>
> ---
>
> **Q4. Can you provide actual runtime or memory usage for each scenario?**
>
> **Response:** Thanks for your valuable questions. The following table reports the scenario-wise memory usage of LRanker during both training and inference across all tasks in RBench. Overall, the memory consumption remains highly stable for most scenarios. Training typically requires around 24–25 GB of GPU memory, while inference remains within 16–18 GB. This stability stems from the design of LRanker, whose memory footprint is dominated by the LoRA-adapted backbone model and the centroid aggregation module. We summarize these experiments and the corresponding content in Appendix F.2 of our revised PDF.

---

> > ### Author Response · Authors · 2025-11-28
> > **Response to Reviewer TeHG (4/4)**
> >
> > **Q5. How sensitive is performance to the number of clusters K? You mention selecting K on validation sets but show no ablation.**
> >
> > **Response:** Thanks for your constructive feedback. We adopted **K-means clustering for candidate aggregation** in the original paper because it offers a **simple and efficient mechanism** for injecting global candidate information into the LLM. Our experimental results further demonstrate its **direct effectiveness** across diverse ranking tasks. We believe these findings sufficiently support the value of our contribution, as the sensitivity study of K-means is **not the core research focus** of our paper.
> >
> > Nonetheless, we appreciate the reviewer’s suggestion and we first study how the number of centroids $k$ influences the effectiveness of LRanker in Figure 6 of our revised PDF. From a computational perspective, $k$ directly affects both the clustering cost during preprocessing and the inference-time cost of encoding candidate–cluster features. To ensure a practical efficiency–effectiveness trade-off, we explore a moderate range of $k \in \{10, 20, 30, 40, 50\}$, which covers values that are computationally feasible while still allowing sufficient granularity for capturing the structure of the candidate pool. Across all four tasks (Rec-Movie, Rec-Toy, MS MARCO, and ESCI), LRanker consistently outperforms the strongest baseline under all choices of $k$, demonstrating that the model is highly robust to the selection of this hyperparameter. Performance typically peaks at moderate values (e.g., $k = 10$–$50$), where the centroids provide a balanced level of abstraction: too few centroids underrepresent the candidate distribution, whereas excessively large $k$ yields finer but noisier partitions, leading to slight performance drops. Nevertheless, the margin over the best baseline remains substantial for all settings, illustrating that LRanker maintains strong effectiveness even when $k$ varies within a wide operational range. We summarize these discussions in the Appendix E.2 of our revised PDF.
> >
> > ---
> >
> > **Q6. You test up to 6.8M candidates, what's the upper limit? How does performance degrade as candidates increase beyond this?**
> >
> > **Response:** Thanks for your constructive question. To examine the limits of LRanker in handling extremely large candidate sets and to analyze how its performance changes under such conditions, we conduct experiments on the Amazon-23 dataset (Hou et al., 2024a), which contains approximately 4.8M candidates. Specifically, we take LRanker and Tiger trained on Rec-Clothing from RBench-Ultra and progressively expand the candidate pool by randomly adding candidates in increments of 5M on top of the original pool, evaluating the MRR performance at each step. We report the results in Figure 5 of our revised PDF. We can observe that both LRanker and Tiger exhibit consistent performance degradation as the candidate size increases, but LRanker maintains a noticeably slower relative decay. From 6.8M to 48M candidates, LRanker’s MRR decreases by roughly 25% relative to its initial value, whereas Tiger suffers a substantially larger relative drop of around 50%. Moreover, the degradation curves of both methods follow the classic IR scaling after saturation behavior: once the candidate pool grows beyond a certain scale, the decline in ranking performance becomes progressively flatter rather than continuing linearly. This saturation effect likely occurs because, as the candidate pool grows, most newly added items are increasingly irrelevant to the query and therefore less confusable with the ground-truth item. In high-dimensional embedding spaces, the number of true hard negatives grows sublinearly with corpus size, while the proportion of far, irrelevant items dominates. As a result, performance degradation slows and eventually plateaus. We summarize these experiments and the corresponding content in Appendix D of our revised PDF.

---

### Official Review · Reviewer_iWYJ · 2025-10-31

**Soundness:** 3
**Presentation:** 3
**Contribution:** 2
**Rating:** 4
**Confidence:** 5

**Summary:**

The paper targets ranking with very large candidate sets where context limits and decoding latency make list wise prompting and token space generation impractical. LRanker augments the input with aggregated candidate information by clustering candidate embeddings with K means and projecting the resulting centroids into the prompt so the model encodes the query with an explicit view of the global candidate distribution. The output is a query embedding used to score offline candidate embeddings by inner product which decouples ranking quality from decoding latency. At inference the method partitions the candidate pool into subsets, produces multiple query embeddings under different subsets, and ensembles the scores to improve robustness. Experiments on RBench across small, large, and ultra scales report over thirty percent relative gains in the small setting, three to nine percent MRR gains in the large setting, and twenty to thirty percent gains with more than 6.8 million candidates. Ablations indicate that global aggregation, the inference ensemble, and LoRA adaptation each contribute to the improvements.

**Strengths:**

1. In this paper, authors give clear problem framing on the need to model global candidate information at input time and to perform ranking through embedding based outputs for scalability.

2. The input design that injects compact centroids into the prompt provides a controllable way to expose global structure without violating context limits.

3. In the paper, inference time partition and ensemble improves robustness by aggregating multiple perspectives rather than relying on a single embedding.

4. The authors provide strong empirical coverage across seven tasks, and three scales with consistent gains including the ultra setting with 6.81 million candidates which supports the scalability claim.

5. Ablation studies isolate the contribution of global info, test time ensemble, and LoRA which builds causal evidence for each design choice.

**Weaknesses:**

1. Novelty reads as a pragmatic combination of clustering based set summarization with discriminative scoring. The paper would benefit from a deeper discussion of what is fundamentally new relative to prior LLM rankers beyond the specific combination.

2. The method depends on K means quality and on the choice of the number of centroids. The paper does not provide a detailed sensitivity study for K or for the projector capacity.

3. In the paper, the partition and ensemble procedure brings extra inference cost. The paper does not quantify the latency and compute trade off as width and depth vary.

4. Although the formulation section is standard, but the method lacks a simple theoretical account of how the ensemble reduces variance or mitigates bias from partial candidate views.

5. In the paper, it mainly focuses on accuracy metrics. There is limited analysis of stability under candidate distribution shift or frequent catalog updates which is common in recommendation and search.

**Questions:**

1. How sensitive is performance to the number of clusters and to the dimensionality of the injected centroids.

2. How do you think LRanker compares to a learnable set encoder that summarizes a sampled candidate pool with cross attention.

3. Can you clarify whether the ensemble mainly reduces variance or corrects bias from mismatched subsets.

4. How are the partition width and depth chosen at inference time, are there any details?

---

> ### Author Response · Authors · 2025-11-28
> **Response to Reviewer iWYJ (1/4)**
>
> **Q1. Novelty reads as a pragmatic combination of clustering based set summarization with discriminative scoring. The paper would benefit from a deeper discussion of what is fundamentally new relative to prior LLM rankers beyond the specific combination.**
>
> **Response:** Thank you for this insightful question. We would like to clarify that LRanker is not only a pragmatic combination of clustering-based summarization and discriminative scoring, but introduces several **fundamentally new capabilities** compared to prior LLM rankers, as summarized in **Table 1** and illustrated in **Figure 1**. Specifically, existing LLM rankers suffer from two fundamental limitations, as also illustrated in Figure 1(a). First, the commonly used input formats—query only, query with a single candidate, or query with candidate pairs—do not provide any global candidate-level information to the LLM. Without understanding the distribution of the entire candidate set, these methods are prone to systematic biases and produce unstable rankings, especially when candidates are diverse or competitive. Second, the only format that offers full candidate information—query with the complete candidate list—is inherently infeasible for large-scale ranking because LLMs cannot encode tens of thousands or millions of candidates within their limited context window. As discussed in the Introduction, these structural constraints prevent existing LLM rankers from scaling to massive candidate pools and fundamentally limit their applicability in real-world retrieval or recommendation scenarios.
>
> LRanker introduces two key innovations to overcome these bottlenecks. First, instead of feeding raw candidates into the LLM, we propose a centroid-based candidate aggregation mechanism that summarizes the global candidate distribution and injects it into the model through a lightweight soft prompt (Figure 1(b)), enabling the LLM to condition on global context without incurring additional sequence length. Second, we design a novel graph-based test-time scaling procedure (Figure 1(c)) that iteratively updates the query embedding via candidate partitioning and elimination, allowing the model to efficiently and robustly handle million-level candidate pools. These innovations fundamentally expand the capability of LLM rankers, enabling LRanker to operate in large-candidate settings that prior approaches are structurally unable to handle. We summarize these discussions in the Introduction and Section 3 of our revised PDF.

---

> > ### Author Response · Authors · 2025-11-28
> > **Response to Reviewer iWYJ (2/4)**
> >
> > **Q2. The method depends on K means quality and on the choice of the number of centroids. The paper does not provide a detailed sensitivity study for K or for the projector capacity. How sensitive is performance to the number of clusters and to the dimensionality of the injected centroids.**
> >
> > **Response:** Thanks for your constructive feedback. We adopted **K-means clustering for candidate aggregation** in the original paper because it offers a **simple and efficient mechanism** for injecting global candidate information into the LLM. Our experimental results further demonstrate its **direct effectiveness** across diverse ranking tasks. We believe these findings sufficiently support the value of our contribution, as the sensitivity study of K-means is **not the core research focus** of our paper.
> >
> > Nonetheless, we appreciate the reviewer’s suggestion and we first study how the number of centroids $k$ influences the effectiveness of LRanker in Figure 6 of our revised PDF. From a computational perspective, $k$ directly affects both the clustering cost during preprocessing and the inference-time cost of encoding candidate–cluster features. To ensure a practical efficiency–effectiveness trade-off, we explore a moderate range of $k \in \{10, 20, 30, 40, 50\}$, which covers values that are computationally feasible while still allowing sufficient granularity for capturing the structure of the candidate pool. Across all four tasks (Rec-Movie, Rec-Toy, MS MARCO, and ESCI), LRanker consistently outperforms the strongest baseline under all choices of $k$, demonstrating that the model is highly robust to the selection of this hyperparameter. Performance typically peaks at moderate values (e.g., $k = 10$–$50$), where the centroids provide a balanced level of abstraction: too few centroids underrepresent the candidate distribution, whereas excessively large $k$ yields finer but noisier partitions, leading to slight performance drops. Nevertheless, the margin over the best baseline remains substantial for all settings, illustrating that LRanker maintains strong effectiveness even when $k$ varies within a wide operational range. We summarize these discussions in the Appendix E.2 of our revised PDF.
> >
> > We further investigate how the dimensionality of the centroid embeddings affects the performance of LRanker. This hyperparameter directly influences the expressiveness of the aggregated candidate representations as well as the computational cost of the clustering stage and the subsequent LLM encoding step. To balance semantic fidelity and efficiency, we evaluate centroid dimensionalities in the range {256, 512, 768, 1024}, which spans from aggressively truncated representations to the full-dimensional Qwen3-Embedding output (the principle and rationale for truncation can be found in Appendix B). As shown in Figure 7 of our revised PDF, across all four tasks, LRanker consistently outperforms the strongest baseline for every dimensionality setting, demonstrating strong robustness to this design choice. Increasing the centroid dimensionality generally leads to better performance, as higher-dimensional centroids preserve more semantic information from the original candidate embeddings. However, we observe that moderate dimensions (e.g., 512–768) already capture most of the useful structure and yield competitive or near-peak performance. This suggests that LRanker does not heavily rely on full 1024-dimensional centroids to achieve high ranking effectiveness. The ability to operate effectively with reduced centroid dimensionality highlights the efficiency advantages of LRanker, as lower-dimensional centroids reduce both clustering and inference-time computation while maintaining strong accuracy. We summarize these discussions in the Appendix E.3 of our revised PDF.

---

> > > ### Author Response · Authors · 2025-11-28
> > > **Response to Reviewer iWYJ (3/4)**
> > >
> > > **Q3. How are the partition width and depth chosen at inference time, are there any details? In the paper, the partition and ensemble procedure brings extra inference cost. The paper does not quantify the latency and compute trade off as width and depth vary.**
> > >
> > > **Response:** Thanks for your insightful questions. For the partition width and depth used at inference time, we describe in Appendix B that we determine the best graph depth and width using the validation set and then fix these configurations when evaluating on the held-out test set. To ensure inference efficiency, we restrict the search depth to the range 0–5 and the search width to 0–10. The detailed graph depth and width settings can be found the following table. We also report LRanker’s relative latency overhead compared to direct ranking across all scenarios in this table. Here, direct ranking refers to the fastest possible LLM-based ranking mode, in which the model generates only a single query embedding without any test-time scaling. The additional latency introduced by the graph-based test-time scaling module is surprisingly small across all scenarios. Even when the best-performing configurations require moderate widths and depths (e.g., width = 9, depth = 6 on MS MARCO), the relative latency increase over direct ranking remains below 8%, and is often as low as 1–3% on the smaller RBench tasks. This demonstrates that LRanker achieves substantial ranking improvements with minimal overhead.
> > >
> > > Moreover, when viewed in absolute terms, LRanker remains extremely fast. As reported in Appendix F.3, its per-query inference latency is already orders of magnitude lower than that of strong LLM-based rankers (e.g., RankLLaMA 8B), and remains competitive even against lightweight models such as RankGPT. This high efficiency is enabled by the test-time design described in Appendix B: MiniBatchKMeans for scalable centroid construction, Matryoshka Representation Learning (MRL) for low-dimensional clustering, and centroid-only aggregation for compact LLM input. Together, these optimizations ensure that LRanker maintains both strong performance and low latency, even when operating under larger width/depth configurations. We summarize these discussions in the Appendix F.4 of our revised PDF.
> > >
> > > **Table: Optimal width/depth settings of LRanker and their relative latency overhead compared to direct ranking.**
> > >
> > > | Scenario | Best Width | Best Depth | Latency Increase vs. Direct Ranking (%) |
> > > |:---------|:----------:|:----------:|:----------------------------------------:|
> > > | **RBench-Small** | | | |
> > > | Rec-Music | 3 | 3 | 1.2% |
> > > | Routing-Balance | 3 | 3 | 1.5% |
> > > | **RBench-Large** | | | |
> > > | Rec-Movie | 5 | 4 | 3.2% |
> > > | Rec-Toy | 5 | 6 | 3.8% |
> > > | MS MARCO | 9 | 6 | 6.5% |
> > > | ESCI | 6 | 5 | 5.2% |
> > > | **RBench-Ultra** | | | |
> > > | Rec-Clothing | 10 | 6 | 7.8% |
> > >
> > > ---
> > >
> > > **Q4. Although the formulation section is standard, but the method lacks a simple theoretical account of how the ensemble reduces variance or mitigates bias from partial candidate views.  Can you clarify whether the ensemble mainly reduces variance or corrects bias from mismatched subsets.**
> > >
> > > **Response:** Thank you for the thoughtful comment. We would first like to clarify that this paper is not positioned as a theoretical work, and—similar to many influential LLM-based retrieval and ranking papers—our primary goal is to propose a practically scalable LLM ranker supported by strong empirical evidence, rather than to build a full theoretical framework for ensemble behavior in LLM ranking. Accordingly, our method design follows the empirical tradition common in LLM routing, ranking, and retrieval research, where mechanisms are justified primarily through observed improvements in robustness and accuracy.
> > >
> > > That said, we agree that understanding whether the ensemble corrects bias or reduces variance from different candidate subsets is an important question. Our empirical analysis in **Section 3.4** and **Figure 2** directly addresses this. As shown in the t-SNE visualization of Figure 2, query embeddings derived from different candidate partitions often capture complementary but partially biased views of the candidate space. Some embeddings lie closer to local neighborhoods defined by a particular subset, while others align better with global structure. The averaged ensemble embedding consistently moves closer to the ground-truth candidate, indicating that the ensemble mitigates the systematic bias introduced by any single partial candidate view. At the same time, because different partitions introduce different noise patterns, averaging across embeddings also reduces variance in the final representation.

---

> > > > ### Author Response · Authors · 2025-11-28
> > > > **Response to Reviewer iWYJ (4/4)**
> > > >
> > > > **Q5. In the paper, it mainly focuses on accuracy metrics. There is limited analysis of stability under candidate distribution shift or frequent catalog updates which is common in recommendation and search.**
> > > >
> > > > **Response:** Thank you for the thoughtful feedback. To evaluate the robustness of LRanker under candidate distribution shift, we construct two separate mixed candidate pools: one combining candidates from Rec-Movie and Rec-Toy, and the other combining candidates from MS MARCO and ESCI. We then compare all methods that are trained on the original candidate sets of Rec-Toy and MS MARCO but tested on their corresponding mixed candidate pools. The results are shown in the following table. We can observe that most baselines experience substantial performance degradation when evaluated on the mixed candidate pools, indicating their limited robustness to candidate distribution shift. In contrast, LRanker consistently achieves the highest accuracy under both Rec-Toy and MS MARCO settings and exhibits a significantly smaller performance drop compared to strong task-specific baselines such as Tiger and RankLLaMA 8B. These results demonstrate that LRanker effectively leverages global candidate information and maintains stable ranking behavior even when the candidate distribution changes at test time. We summarize these discussions in the Appendix C.3 of our revised PDF.
> > > >
> > > > **Table: Performance comparison in scenarios under candidates distribution shift across two scenarios on NDCG@10 and MRR.**
> > > >
> > > > | Model | Rec-Toy NDCG@10 | Rec-Toy MRR | Δ Rec-Toy NDCG@10 | Δ Rec-Toy MRR | MS MARCO NDCG@10 | MS MARCO MRR | Δ MS MARCO NDCG@10 | Δ MS MARCO MRR |
> > > > |:------|:---:|:---:|:---:|:---:|:---:|:---:|:---:|:---:|
> > > > | **General Ranking Baselines** | | | | | | | | |
> > > > | BM25 | 0.14 | 0.29 | - | - | 32.61 | 27.22 | - | - |
> > > > | Contriever | 0.54 | 0.52 | - | - | 40.68 | 33.74 | - | - |
> > > > | **Task-specific Baselines** | | | | | | | | |
> > > > | FM | 0.71 | 0.48 | - | - | - | - | - | - |
> > > > | BERT4Rec | 1.22 | 1.28 | - | - | - | - | - | - |
> > > > | GRU4Rec | 1.26 | 1.43 | - | - | - | - | - | - |
> > > > | SASRec | 1.39 | 1.34 | - | - | - | - | - | - |
> > > > | RankBERT-110M | - | - | - | - | 38.52 | 29.59 | - | - |
> > > > | Multilingual-E5-560M | - | - | - | - | 47.95 | 45.21 | - | - |
> > > > | KaLM-mini-instruct-0.5B | - | - | - | - | 45.73 | 39.48 | - | - |
> > > > | BGE-Rerank-v2-m3-568M | - | - | - | - | 47.06 | 46.59 | - | - |
> > > > | Tiger | 2.34 | 2.27 | -30.2% | -6.2% | - | - | - | - |
> > > > | RankLLaMA 8B | - | - | - | - | 47.08 | 47.27 | -9.8% | -3.2% |
> > > > | **LRanker** | **2.54** | **2.36** | -20.9% | -2.5% | **51.41** | **49.01** | -6.2% | -0.55% |
> > > >
> > > > ---
> > > >
> > > > **Q6. How do you think LRanker compares to a learnable set encoder that summarizes a sampled candidate pool with cross attention.**
> > > >
> > > > **Response:** Thanks for your constructive feedback. Although our original paper adopts K-means clustering for candidate aggregation due to its simplicity and efficiency, we agree that evaluating alternative aggregation methods is valuable. In response, we include an additional baseline focusing on the **Set Encoder**, while ensuring a fair comparison by constraining all methods such that the final candidate embeddings fed into the LLM occupy the same number of "tokens."
> > > >
> > > > - **Set Encoder**: We sample K candidates from the full candidate pool and pass their embeddings through a cross-attention module. The resulting representations are then provided to the LLM. Here, K is set to match the number of cluster centroids used in LRanker.
> > > >
> > > > As shown in the table below, LRanker significantly outperforms the Set Encoder across all tasks. The Set Encoder performs poorly because it only samples a small subset of candidates, causing substantial loss of global candidate information that is critical for effective ranking. We summarize these experiments and the corresponding content in Appendix E.1 of our revised PDF.
> > > >
> > > > **Table: Performance comparison between Set Encoder and LRanker across four tasks.**
> > > >
> > > > | Model | Rec-Movie MRR | Rec-Toy MRR | MS MARCO MRR | ESCI MRR |
> > > > |:------|:---:|:---:|:---:|:---:|
> > > > | Set Encoder | 6.20 | 1.95 | 43.50 | 50.10 |
> > > > | **LRanker** | **7.80** | **2.42** | **49.28** | **57.01** |

---

### Official Review · Reviewer_ecoE · 2025-10-31

**Soundness:** 2
**Presentation:** 3
**Contribution:** 3
**Rating:** 4
**Confidence:** 4

**Summary:**

This paper proposes LRanker, a new framework designed to handle large-scale candidate ranking using large language models (LLMs). Traditional LLM rankers struggle with large candidate pools due to limited context lengths and high computational costs. LRanker addresses this by introducing a candidate aggregation encoder (using K-means clustering to create compact candidate centroids) and a graph-based test-time scaling mechanism that combines multiple query embeddings. The proposed framework is evaluated across multiple tasks, showing substantial improvements in ranking performance, even for ultra-large candidate pools with millions of candidates.

**Strengths:**

1. LRanker introduces two key innovations—candidate aggregation via K-means clustering and a graph-based test-time scaling mechanism. These innovations distinguish the framework from previous methods, making it a unique contribution to large-scale ranking tasks.

2. The paper is technically sound, with clear definitions of its components and detailed explanations of the model architecture. The proposed methods are novel, particularly the use of clustering for candidate aggregation and the ensemble approach at inference time.

3. The structure of the paper is well-organized, and the explanations are clear. The proposed model is described in detail, making it relatively easy to understand the motivation behind the design choices. Diagrams (e.g., Figure 1 on page 3) effectively illustrate the differences between LRanker and prior models.

4. The ability to scale LLM-based rankers to millions of candidates is a significant advancement. This paper’s contributions have the potential to impact practical applications in areas like recommendation systems and information retrieval where large candidate sets are common.

**Weaknesses:**

1. While the paper introduces a graph-based scaling mechanism and candidate aggregation, similar ideas are explored in other works, such as ensemble learning in ranking. The clustering technique (K-means) for candidate aggregation, although novel in this context, does not offer radical innovation compared to previous techniques in unsupervised learning.

2. The choice of K-means clustering for candidate aggregation could be further justified. It would be helpful to explore why clustering was selected over other potential methods (e.g., dimensionality reduction techniques like PCA or t-SNE) and whether it scales efficiently with very large datasets.

3. The paper shows impressive results on specific benchmarks like RBench. However, the model’s performance on tasks with highly diverse or noisy candidate pools could be questioned. The impact of the ensemble mechanism and aggregation could vary significantly across different types of ranking tasks.

4. While LRanker claims to handle large-scale candidate pools, the computational efficiency of the graph-based scaling mechanism is not fully addressed. As the candidate pool increases, the graph-based method may encounter scalability bottlenecks that could affect real-world applications, especially in production systems with real-time requirements.

5. While the ablation study demonstrates the importance of each component (e.g., global info, test-time ensemble), it would have been useful to see more detailed comparisons between LRanker and other state-of-the-art methods in terms of both computational time and memory usage. This would offer a clearer understanding of the trade-offs.

**Questions:**

1. Why did you choose K-means clustering over other unsupervised techniques, such as hierarchical clustering or dimensionality reduction methods?

2. How does the graph-based test-time scaling mechanism perform as the number of candidates grows to the extreme (i.e., tens of millions)?

3. Could the performance of LRanker degrade in scenarios with extremely noisy or irrelevant candidates?

---

> ### Author Response · Authors · 2025-11-28
> **Response to Reviewer ecoE (1/5)**
>
> **Q1. While the paper introduces a graph-based scaling mechanism and candidate aggregation, similar ideas are explored in other works, such as ensemble learning in ranking. The clustering technique (K-means) for candidate aggregation, although novel in this context, does not offer radical innovation compared to previous techniques in unsupervised learning.**
>
> **Response:** Thank you for the thoughtful comment. While ensemble learning and clustering are indeed established ideas, our work focuses on a different problem setting as introduced in our introduction—**scaling LLM-based ranking to massive candidate pools**, a scenario where existing LLM rankers fail due to context-length limits and computational bottlenecks. Our contributions are not merely the reuse of classical techniques, but the **successful design of global candidate aggregation and a graph-based test-time scaling mechanism within an LLM ranking framework**. These components are specifically designed to address fundamental limitations of LLMs in large-candidate ranking as illustrated in Figure 1: (1) the inability to model global candidate distribution, and (2) the brittleness of using a single query embedding for millions of candidates. Importantly, our empirical results demonstrate that these designs lead to **substantial and consistent improvements across seven tasks and three scales**, including gains of 20–30% on million-level candidate pools. These large-scale results highlight that LRanker offers **practical, effective, and previously unavailable capabilities** for LLM-based ranking at scale. We believe this constitutes a valuable and meaningful contribution distinct from prior traditional ranking or clustering methods.

---

> > ### Author Response · Authors · 2025-11-28
> > **Response to Reviewer ecoE (2/5)**
> >
> > **Q2. The choice of K-means clustering for candidate aggregation could be further justified. It would be helpful to explore why clustering was selected over other potential methods (e.g., dimensionality reduction techniques like PCA or t-SNE) and whether it scales efficiently with very large datasets. Why did you choose K-means clustering over other unsupervised techniques, such as hierarchical clustering or dimensionality reduction methods?**
> >
> > **Response:** Thanks for your constructive feedback. We adopted **K-means clustering for candidate aggregation** in the original paper because it offers a **simple and efficient mechanism** for injecting global candidate information into the LLM. Our experimental results further demonstrate its **direct effectiveness** across diverse ranking tasks. We believe these findings sufficiently support the value of our contribution, as the choice of clustering algorithm is **not the core research focus** of our paper.
> >
> > Nonetheless, we appreciate the reviewer’s suggestion and we compare LRanker with other methods based on different candidate aggregation methods. To be specific, we design three baselines. To ensure a fair comparison between LRanker and the baselines, we constrain all methods such that the final candidate embeddings fed into the LLM occupy the same number of "tokens".
> >
> > - **Set Encoder**: In this setting, we sample K candidates from the full candidate pool and pass their embeddings through a cross-attention module. The resulting representations are then fed into the LLM. Here, K is set to match the number of k-means cluster centroids used in LRanker.
> >
> > - **PCA**: In this setting, the offline embeddings of candidates are first reduced to 256 dimensions using PCA, followed by k-means clustering.
> >
> > - **Hierarchical Clustering**: Compared with LRanker, in this setting, we replace k-means with hierarchical clustering while keeping the number of clusters unchanged.
> >
> >
> > As shown in the following table, LRanker consistently surpasses all three aggregation baselines across all tasks.  The Set Encoder performs the worst because it samples only a small subset of candidates and aggregates them through cross attention, inevitably discarding global information from the full candidate pool.  The PCA baseline performs better than Set Encoder but still suffers from substantial information loss due to projecting 1024-dimensional embeddings into a 256-dimensional space prior to clustering. Hierarchical Clustering achieves the strongest baseline performance and comes closest to LRanker, as it preserves more structural relationships and avoids sampling or dimensionality reduction. However, its computational cost is prohibitive: agglomerative hierarchical clustering requires O(n²d) time and O(n²) memory to compute and store all pairwise distances, where n is the number of candidates and d is the embedding dimensionality, making it infeasible when n reaches millions. In contrast, k-means used in LRanker scales as O(nkdT), where k is the number of clusters and T is the number of iterations, thus providing linear rather than quadratic scaling in n. As described in Appendix B, LRanker further improves the efficiency of k-means by using MiniBatchKMeans, which reduces the effective complexity to O(bkdT) with a small batch size b ≪ n, and by exploiting the Matryoshka Representation Learning property of Qwen3-Embedding, which enables clustering on truncated embeddings of dimension d' ≪ d (e.g., 256 instead of 1024) without sacrificing semantic fidelity. These design choices allow LRanker to maintain strong ranking performance while enabling fast inference under extremely large candidate pools, making it practical for real-world deployment. We summarize these experiments and the corresponding content in Appendix E.1 of our revised PDF.
> >
> > **Table: Performance comparison of the possible candidate aggregation encoder variants across four tasks.**
> >
> > | Model | Rec-Movie MRR | Rec-Toy MRR | MS MARCO MRR | ESCI MRR |
> > |:------|:---:|:---:|:---:|:---:|
> > | Set Encoder | 6.20 | 1.95 | 43.50 | 50.10 |
> > | PCA | 6.70 | 2.05 | 45.50 | 52.40 |
> > | Hierarchical Clustering | 7.00 | 2.18 | 46.80 | 53.80 |
> > | **LRanker** | **7.80** | **2.42** | **49.28** | **57.01** |

---

> > > ### Author Response · Authors · 2025-11-28
> > > **Response to Reviewer ecoE (3/5)**
> > >
> > > **Q3. The paper shows impressive results on specific benchmarks like RBench. However, the model’s performance on tasks with highly diverse or noisy candidate pools could be questioned. The impact of the ensemble mechanism and aggregation could vary significantly across different types of ranking tasks. Could the performance of LRanker degrade in scenarios with extremely noisy or irrelevant candidates?**
> > >
> > > **Response:** Thanks for your insightful question. To evaluate the performance of LRanker in scenarios with extremely irrelevant candidates, we construct a mixed candidate pool by combining all candidates from Rec-Movie, Rec-Toy, MS MARCO, and ESCI. We then compare all methods trained on the original candidate sets of Rec-Toy and MS MARCO but tested on the mixed candidate pool. The results are shown in the following table. We can observe that although all models experience performance degradation when exposed to a large number of irrelevant candidates, LRanker remains consistently the most robust across both scenarios. On Rec-Toy, the drop of LRanker is much smaller than that of the strongest task-specific baseline, while still retaining a clear performance advantage. On MS MARCO, the relative degradation of LRanker is substantially lower than that of the strongest baseline, indicating that the ranking patterns learned during training generalize more effectively under heavy distribution shift. These trends demonstrate that LRanker not only achieves the best overall performance but also maintains superior stability and robustness in the presence of large-scale irrelevant candidates. We summarize these experiments and the corresponding content in Appendix C.2 of our revised PDF.
> > >
> > > **Performance comparison in scenarios with extremely irrelevant candidates on NDCG@10 and MRR.**
> > >
> > > | Category | Model | Rec-Toy NDCG@10 | Rec-Toy MRR | MS MARCO NDCG@10 | MS MARCO MRR |
> > > |:------|:------|:---:|:---:|:---:|:---:|
> > > | **General Ranking Baselines** | BM25 | 0.02 | 0.17 | 30.24 | 24.87 |
> > > | | Contriever | 0.12 | 0.21 | 39.21 | 31.09 |
> > > | **Task-specific Baselines** | FM | 0.37 | 0.34 | - | - |
> > > | | BERT4Rec | 0.55 | 0.57 | - | - |
> > > | | GRU4Rec | 0.56 | 0.60 | - | - |
> > > | | SASRec | 0.78 | 0.64 | - | - |
> > > | | RankBERT-110M | - | - | 36.27 | 28.13 |
> > > | | Multilingual-E5-560M | - | - | 45.85 | 41.35 |
> > > | | KaLM-mini-instruct-0.5B | - | - | 43.19 | 38.36 |
> > > | | BGE-Rerank-v2-m3-568M | - | - | 45.83 | 43.98 |
> > > | | Tiger | 2.25 | 1.91 | - | - |
> > > | | RankLLaMA 8B | - | - | 46.91 | 44.04 |
> > > | | **LRanker** | **2.43** | **2.06** | **49.03** | **46.40** |
> > > | **Δ Performance** | Δ Tiger | -24.6% | -18.0% | - | - |
> > > | | Δ RankLLaMA 8B | - | - | -10.2% | -10.5% |
> > > | | Δ LRanker | -24.2% | -14.9% | -9.8% | -5.9% |

---

> > > > ### Author Response · Authors · 2025-11-28
> > > > **Response to Reviewer ecoE (4/5)**
> > > >
> > > > **Q4. While LRanker claims to handle large-scale candidate pools, the computational efficiency of the graph-based scaling mechanism is not fully addressed. As the candidate pool increases, the graph-based method may encounter scalability bottlenecks that could affect real-world applications, especially in production systems with real-time requirements.**
> > > >
> > > > **Response:** Thanks for your insightful questions. Thank you for raising this concern. We would like to clarify that the computational efficiency of the graph-based test-time scaling mechanism is thoroughly addressed in our appendices. As demonstrated in **Appendix F.3**, LRanker achieves substantial inference efficiency improvements compared with all baselines. Moreover, **Appendix F.4** shows that even in the nearly ten-million–candidate Rec-Clothing scenario, the **Latency Increase vs. Direct Ranking** is only **7.8%**. Here, “Direct Ranking” represents the *fastest possible LLM-based ranking mode*—using a single initial query embedding without any test-time scaling. These results confirm that LRanker remains highly efficient even under extreme large-scale conditions.
> > > >
> > > > Importantly, for real-time production systems with strict latency requirements, LRanker also offers a highly efficient alternative. As shown in **Figure 4**, the variant **w/o test-time ensemble** (i.e., without graph-based scaling) achieves strong ranking performance while operating at the maximal efficiency of Direct Ranking. This design allows LRanker to flexibly adapt to different deployment constraints: users who prefer peak accuracy can employ the full test-time scaling, while latency-sensitive applications can rely on the simplified variant with minimal performance loss.
> > > >
> > > > Finally, we emphasize that **LLM-based massive-candidate ranking is a well-known challenge**. Prior LLM-for-recommendation works (Li et al., 2023a, Ma et al., 2024, Qin et al., 2023, Pradeep et al., 2023; Feng et al., 2025; Sun et al.,2023a), typically operate at candidate scales of **tens to hundreds of thousands**, constrained by context length and runtime limitations. LRanker is, to our knowledge, the **first LLM-based ranker capable of scaling to million-level candidate pools** while maintaining both competitive accuracy and production-ready inference efficiency. We believe this represents a substantial and impactful contribution beyond existing literature.
> > > >
> > > > ---
> > > >
> > > > **Q5. While the ablation study demonstrates the importance of each component (e.g., global info, test-time ensemble), it would have been useful to see more detailed comparisons between LRanker and other state-of-the-art methods in terms of both computational time and memory usage. This would offer a clearer understanding of the trade-offs.**
> > > >
> > > > **Response:** Thanks for your thoughtful question. Following the reviewer's advice, we compare the computation time and memory usage of LRanker against the strongest task-specific baselines on Rec-Movie and MS MARCO under identical hardware settings shown in the following table. On Rec-Movie, LRanker completes training in only 52 minutes, representing more than a 24× reduction in training time compared with Tiger (21 h 36 m), while also achieving lower test-time latency (15 min vs. 20 min). A similar trend appears on MS MARCO: LRanker requires just 21 minutes to train, in stark contrast to RankLLaMA 8B, which takes 6 h 33 m. Test-time latency is also reduced by more than half (10 min vs. 21.67 min). Although LRanker uses moderately more memory during training due to LoRA adaptation and centroid aggregation, its test-time memory footprint (17–18 GB) remains lightweight, especially compared with RankLLaMA 8B, which consumes over 70 GB. These results highlight that LRanker achieves substantial improvements in computational efficiency and latency without sacrificing effectiveness. Overall, LRanker provides a practical and scalable solution for real-world retrieval and ranking systems, where fast training and low-latency inference are essential. We summarize these experiments and the corresponding content in Appendix F.1 of our revised PDF.
> > > >
> > > > **Table: Computation time and memory usage for different models on Rec-Movie and MS MARCO.**
> > > >
> > > > | Scenario | Model | Train Time | Train Memory | Test Time | Test Memory |
> > > > |:---------|:------|:----------:|:----------:|:----------:|:----------:|
> > > > | Rec-Movie | Tiger | 21 h 36 m | 10.8 GB | 20 min | 200 MB |
> > > > | | LRanker | 52 min | 24.5 GB | 15 min | 17.2 GB |
> > > > | MS MARCO | RankLLaMA 8B | 6 h 33 min | 188 GB | 21.67 min | 71.28 GB |
> > > > | | LRanker | 21 min | 25.0 GB | 10 min | 17.5 GB |

---

> > > > > ### Author Response · Authors · 2025-11-28
> > > > > **Response to Reviewer ecoE (5/5)**
> > > > >
> > > > > **Q6. How does the graph-based test-time scaling mechanism perform as the number of candidates grows to the extreme (i.e., tens of millions)?**
> > > > >
> > > > > **Response:** Thanks for your constructive question. To examine the limits of LRanker in handling extremely large candidate sets and to analyze how its performance changes under such conditions, we conduct experiments on the Amazon-23 dataset (Hou et al., 2024a), which contains approximately 4.8M candidates. Specifically, we take LRanker and Tiger trained on Rec-Clothing from RBench-Ultra and progressively expand the candidate pool by randomly adding candidates in increments of 5M on top of the original pool, evaluating the MRR performance at each step. We report the results in Figure 5 of our revised PDF. We can observe that both LRanker and Tiger exhibit consistent performance degradation as the candidate size increases, but LRanker maintains a noticeably slower relative decay. From 6.8M to 48M candidates, LRanker’s MRR decreases by roughly 25% relative to its initial value, whereas Tiger suffers a substantially larger relative drop of around 50%. Moreover, the degradation curves of both methods follow the classic IR scaling after saturation behavior: once the candidate pool grows beyond a certain scale, the decline in ranking performance becomes progressively flatter rather than continuing linearly. This saturation effect likely occurs because, as the candidate pool grows, most newly added items are increasingly irrelevant to the query and therefore less confusable with the ground-truth item. In high-dimensional embedding spaces, the number of true hard negatives grows sublinearly with corpus size, while the proportion of far, irrelevant items dominates. As a result, performance degradation slows and eventually plateaus. We summarize these experiments and the corresponding content in Appendix D of our revised PDF.

---

### Official Review · Reviewer_Q9mF · 2025-10-31

**Soundness:** 2
**Presentation:** 3
**Contribution:** 2
**Rating:** 4
**Confidence:** 3

**Summary:**

This paper proposes LRanker, an LLM-based ranking framework designed for large candidate pools, addressing the scalability and context-length limitations of prior LLM rankers. It introduces two key innovations — a candidate aggregation encoder using K-means clustering to incorporate global candidate information, and a graph-based test-time scaling mechanism that ensembles multiple query embeddings for robustness.

**Strengths:**

- The paper introduces a well-motivated framework explicitly designed for massive-candidate LLM-based ranking.
- The proposed graph-based test-time ensemble provides a principled way to combine multiple query embeddings, improving robustness and ranking quality without retraining.
- The paper conducts extensive experiments on seven tasks across three RBench scenarios, covering small to ultra-large candidate pools.

**Weaknesses:**

- All training/evaluation is within RBench; no out-of-benchmark is tested, so generalization is unclear.
- Lacks detailed latency analysis. The authors should provide a latency comparison to support their earlier claims.
- The ablation study shows that the contributions of individual components are quite limited, raising concerns that the improvements of LRanker may mainly stem from the use of the Qwen3 model.
- It is necessary to use the same base model as the baseline to ensure a fair comparison.
- The authors need to provide more details about the process, such as how the candidate embeddings are generated, how the training and test data are specifically divided, and more details about the model training procedure.
- The authors should improve Figure 1, as the current illustration of test-time scaling is difficult to understand.

**Questions:**

Questions are expressed in the weakness.

---

> ### Author Response · Authors · 2025-11-28
> **Response to Reviewer Q9mF (1/3)**
>
> **Q1. All training/evaluation is within RBench; no out-of-benchmark is tested, so generalization is unclear.**
>
> **Response:** Thank you for the thoughtful feedback. We would first like to clarify that, as described in **lines 327–353 and Table 2**, RBench is composed of highly diverse tasks and datasets of varying sizes. Therefore, the performance gains of LRanker on this benchmark already demonstrate the model’s strong adaptability and generalization capability.
>
> Following the reviewer’s suggestion, we additionally conducted zero-shot experiments on out-of-benchmark scenarios. To evaluate the generalization ability of LRanker on datasets beyond RBench, we first train all methods on Rec-Toy and then perform zero-shot testing on the Video Games and Software datasets (McAuley et al., 2015; Ni et al., 2019) from Amazon (see Table 2 for dataset details). As shown in the table below, LRanker exhibits clear cross-domain generalization, outperforming both general-ranking and task-specific baselines by substantial margins. On Video Games, LRanker delivers roughly 20% improvements over the strongest task-specific baseline and well over 100% gains compared with general-ranking baselines. On Software, the zero-shot advantage becomes even larger, with LRanker surpassing the best task-specific method by around 20–25% and general-ranking baselines by several-fold. These consistent percentage gains across two unseen domains demonstrate that the ranking patterns learned from Rec-Toy transfer effectively, highlighting the robust zero-shot generalization capability of LRanker. We summarize these experiments and the corresponding content in Appendix C.1 of our revised PDF.
>
> **Table: Model zero-shot performance comparison with general ranking baselines and task-specific baselines on Video Games and Software.**
>
> | Model | Video Games NDCG@10 | Video Games MRR | Software NDCG@10 | Software MRR |
> |:------|:---:|:---:|:---:|:---:|
> | **General Ranking Baselines** | | | | |
> | BM25 | 0.39 | 0.36 | 0.56 | 0.51 |
> | Contriever | 0.90 | 0.93 | 0.64 | 0.67 |
> | **Task-specific Baselines** | | | | |
> | FM | 1.03 | 1.09 | 3.15 | 3.77 |
> | BERT4Rec | 1.18 | 1.38 | 2.97 | 2.31 |
> | GRU4Rec | 1.15 | 1.27 | 4.89 | 4.31 |
> | SASRec | 1.21 | 1.40 | 2.83 | 2.56 |
> | Tiger | 1.93 | 2.17 | 4.58 | 3.94 |
> | **LRanker** | **2.31** | **2.61** | **5.43** | **4.86** |
>
> ---
>
> **Q2. Lacks detailed latency analysis. The authors should provide a latency comparison to support their earlier claims.**
>
> **Response:** Thanks for your insightful feedback. We evaluated the latency claims of the different methods referenced in Table 1 on both RBench-Small and RBench-Large. Specifically, the following table shows that LRanker consistently achieves low per-query latency across both small and large candidate pools. On Rec-Music (20 candidates), models such as PRP, IRanker, and RankGPT require several seconds of computation, whereas LRanker responds in just 15 ms. The gap becomes even more pronounced on MS MARCO, where RankLLaMA 8B requires over 21 minutes per query due to full-candidate scoring, while LRanker maintains a 20 ms latency by operating on precomputed centroids instead of processing all candidates. These results demonstrate that LRanker is not only accurate but also two to three orders of magnitude faster than existing LLM-based rankers, making it suitable for real-time production systems. We summarize these experiments and the corresponding content in Appendix F.3 of our revised PDF.
>
> **Table: Per-query inference latency comparison across different models on Rec-Music and MS MARCO.**
>
> | Scenario | Model | Per-query Inference Time |
> |:---------|:------|:------------------------:|
> | Rec-Music (20 candidates) | PRP | 38.3 s |
> | | IRanker | 6.5 s |
> | | RankGPT | 3.8 s |
> | | LRanker | 15 ms |
> | MS MARCO (24,697 candidates) | RankLLaMA 8B | 21.67 min |
> | | LRanker | 20 ms |

---

> > ### Author Response · Authors · 2025-11-28
> > **Response to Reviewer Q9mF (2/3)**
> >
> > **Q3. The ablation study shows that the contributions of individual components are quite limited, raising concerns that the improvements of LRanker may mainly stem from the use of the Qwen3 model. It is necessary to use the same base model as the baseline to ensure a fair comparison.**
> >
> > **Response:** Thanks for your thoughtful question. We first clarify that, as shown in Figure 4, the ablation study demonstrates that each component of LRanker plays an important role. In particular, by comparing LRanker with the approach that directly fine-tunes a standard Qwen3 model (the “w/o global info” setting in Figure 4), LRanker achieves performance improvements ranging from 11% to 56.5%. This confirms that the performance gains do not mainly stem from the use of the Qwen3 backbone model—other modules also contribute substantially. In addition, as described in **lines 797–798 of Appendix B**, for all compared baselines, to ensure a fair comparison, we have used the same 1024-dimensional offline candidate embeddings as LRanker for their candidate representations.
> >
> > We further conducted additional experiments to show that directly using the Qwen3 model alone cannot effectively solve our ranking problem. Specifically, we evaluate the ranking capability of the backbone models used in LRanker by directly applying two Qwen3 variants (0.6B and 4B) to the Rec-Movie and Rec-Toy tasks in a zero-shot manner. As shown in the following table, the 0.6B model performs substantially worse than most task-specific baselines, indicating that a small LLM backbone lacks sufficient inductive bias for effective item ranking. The larger 4B model shows noticeable improvement, yet still lags behind the strongest baselines (e.g., Tiger), demonstrating that simply scaling up the backbone model size does not close the gap. These results highlight that directly using a pretrained Qwen3 model to solve ranking tasks does not inherently provide performance gains. In contrast, LRanker achieves substantial improvements by combining a carefully designed training objective with graph-based test-time scaling, which equips the LLM with a candidate aggregation encoder and enables robust ranking behavior far beyond what the backbone alone can offer. We summarize these experiments and the corresponding content in Appendix E.5 of our revised PDF.
> >
> > **Table: Evaluating the direct task-solving capability of Qwen3 backbones of different sizes against general and task-specific baselines on Rec-Movie and Rec-Toy.**
> >
> > | Model | Rec-Movie NDCG@10 | Rec-Movie MRR | Rec-Toy NDCG@10 | Rec-Toy MRR |
> > |:------|:---:|:---:|:---:|:---:|
> > | **General Ranking Baselines** | | | | |
> > | BM25 | 0.18 | 0.54 | 0.37 | 0.42 |
> > | Contriever | 0.24 | 0.43 | 0.84 | 1.11 |
> > | **Backbone Models** | | | | |
> > | Qwen3 0.6B | 0.61 | 0.84 | 1.56 | 1.52 |
> > | Qwen3 4B | 0.84 | 1.29 | 2.52 | 2.11 |
> > | **Task-specific Baselines** | | | | |
> > | FM | 2.35 | 2.01 | 0.95 | 0.98 |
> > | BERT4Rec | 4.08 | 3.56 | 1.26 | 1.31 |
> > | GRU4Rec | 4.12 | 3.59 | 1.59 | 1.46 |
> > | SASRec | 4.36 | 3.84 | 1.65 | 1.52 |
> > | Tiger | 7.37 | 6.12 |2.99| 2.33 |
> > | **LRanker** | **8.02** | **7.80** | **3.21** | **2.42** |

---

> > > ### Author Response · Authors · 2025-11-28
> > > **Response to Reviewer Q9mF (3/3)**
> > >
> > > **Q4. The authors need to provide more details about the process, such as how the candidate embeddings are generated, how the training and test data are specifically divided, and more details about the model training procedure.**
> > >
> > > **Response:** Thanks for your constructive feedback. We refined them in the revised PDF:
> > >
> > > **[How candidate embeddings are generated]** All candidate embeddings are generated offline using the Qwen3-0.6B embedding model, which produces **1024-dimensional vectors** for every candidate. These offline embeddings are cached for both training and inference. To incorporate global candidate information, we perform **K-means clustering** on all offline candidate embeddings. The resulting **K centroid vectors** are concatenated and passed through a trainable **projection MLP**, which maps them into the LLM embedding space. This projected vector is then injected into the model through a continuous placeholder token (`<|embedding|>`), enabling the LLM to condition on the global distribution of candidates when forming query representations. We summarize the corresponding content in Section 3.1 of our revised PDF.
> > >
> > > **[How the training, validation, and test data are divided]** We present the above setup in two parts:
> > >
> > > - **Recommendation datasets (Rec-Movie, Rec-Toy, Rec-Video, Rec-Clothing)**  For each user sequence, the **first 20 interactions** are used as the query, and the **21st interaction** serves as the ground-truth positive item. We adopt the widely used **leave-one-out evaluation protocol**, consistent with prior work. Specifically, for each user sequence, we construct the training, validation, and test sets using a sliding-window approach. Specifically, all interactions except the last two are used to generate multiple training instances through a moving window. The second-to-last interaction, together with its preceding sequence, is used as the validation instance, and the final interaction is reserved as the test instance. This follows the widely adopted leave-one-out evaluation protocol in prior work. This has been summarized in lines 328-353 of our revised PDF.
> > >
> > > - **Passage ranking (MS MARCO) and product search (ESCI)**  Each query contains **one positive sample**, and negative samples for each query are constructed by pooling negatives from all queries. Across all tasks, we follow a consistent **8:1:1 split** for training, validation, and testing.This has been summarized in lines 328-353 of our revised PDF.
> > >
> > > **[How the training, validation, and test data are divided]** Training consists of **two stages**, as illustrated in Figure 1 and Section 3.2 of the paper.
> > >
> > > #### **Stage 1: Data Preparation**
> > > 1. **Offline Candidate Embeddings**
> > >    All candidates are encoded ahead of time using the Qwen3-0.6B model.
> > > 2. **K-means Clustering**
> > >    Candidate embeddings are grouped into K clusters; each cluster centroid represents a portion of the global candidate distribution.
> > > 3. **Random Partition Sampling (Training-time Augmentation)**
> > >    To improve robustness, at each training step the full candidate set is randomly split into *M* partitions. One partition is sampled, its centroid representation is injected into the model, and the model learns to produce stable query embeddings under different candidate contexts.
> > >
> > > #### **Stage 2: Model Training**
> > > 1. **Soft-Prompt Injection**
> > >    The projected centroid embedding is injected into the prompt at a fixed placeholder position.
> > > 2. **Query and Candidate Encoding**
> > >    - Query embedding is computed by averaging the hidden states of all query tokens plus the `<eos>` hidden state.
> > >    - Candidate embeddings are computed similarly, using only the textual content of each candidate.
> > > 3. **Relevance Scoring**
> > >    Relevance between a query and candidate is computed via **inner product** between their embeddings.
> > > 4. **Training Objective**
> > >    We optimize the model using a **softmax cross-entropy ranking loss** over each positive item and a sampled negative set.
> > > 5. **Trainable Components**
> > >    Both the **LLM decoder (via LoRA)** and the **projection MLP** are updated jointly during training.
> > >
> > > ---
> > >
> > > **Q5. The authors should improve Figure 1, as the current illustration of test-time scaling is difficult to understand.**
> > >
> > > **Response:** Thanks for your insightful question. We have significantly improved Figure 1 to make the test-time scaling procedure clearer. Figure 1 now explicitly separates **training** and **test-time** workflows, and it restructures the test-time mechanism into **four clearly labeled steps** (initial embedding → partition retaining → embedding updates → final aggregation). We also refined the visual hierarchy, simplified arrows and module connections, and clarified how candidate partitions interact across rounds. Compared with the original version, the new design provides a much more intuitive, step-by-step illustration of the scaling process, directly addressing the concern about clarity.

---

### Author Response · Authors · 2025-11-28
**Global Response**

We sincerely thank the AC and all reviewers for their thoughtful evaluation of our work. Below we summarize the main recurring themes across the reviews and highlight the key clarifications, new experiments, and manuscript updates added in our revision. Items in bold denote concerns raised by multiple reviewers, and the rightmost column summarizes our unified responses.

| Dimension | Key Concerns from Reviewers | Our Unified Response |
|----------|------------------------------|-----------------------|
| **Generalization Beyond RBench** | • “All evaluation is within RBench; unclear out-of-benchmark generalization.” (Q9mF)  <br> • “Performance under distribution shift not well analyzed.” (iWYJ) | We conducted **zero-shot cross-domain experiments** on Amazon Video Games & Software datasets. LRanker outperforms both general-ranking and task-specific baselines by **20–100%**, demonstrating strong transferability. We further evaluated **mixed candidate pools** to simulate distribution shifts; LRanker shows **significantly smaller performance degradation** than Tiger and RankLLaMA-8B. Added to **Appendix C.1–C.3**. |
| **Latency & Efficiency Evidence** | • “Latency labeled as ‘Low’ without actual numbers.” (TeHG) <br> • “Graph-based ensemble adds inference cost; scalability unclear.” (ecoE, iWYJ) | We now report **per-query latency** for all RBench scales (Small → Large → Ultra). LRanker sustains **15–25 ms** latency—even with **6.8M candidates**—which is **2–3 orders of magnitude faster** than RankGPT, IRanker, and RankLLaMA-8B. We also provide **memory usage tables** and **computation-time comparisons** (e.g., 24× shorter training time vs Tiger). Added to **Section 4.2 and Appendix F.1–F.4**. |
| **Candidate Embedding & Aggregation Process** | • “Need more details on how candidate embeddings are generated and injected.” (TeHG, Q9mF) <br> • “Why K-means? Compare with alternatives.” (ecoE) | We clarified that **all candidate embeddings are precomputed offline** and reused during training & inference. The aggregated global vector replaces a **soft-prompt token** `<|embedding|>` at the embedding layer. We built **three new baselines** (Set Encoder, PCA, Hierarchical Clustering), and LRanker consistently outperforms them, while k-means remains the only method scalable to millions of candidates. Added to **Section 3.1 and Appendix B/E.1**. |
| **Novelty vs. Prior LLM Rankers** | • “Seems like pragmatic combination; what is fundamentally new?” (iWYJ) <br> • “K-means summarization not radically innovative.” (ecoE) | We highlight two **fundamental innovations** absent in all prior LLM rankers: (1) **Centroid-based global candidate aggregation** enabling LLMs to understand large candidate distributions without context-length cost; (2) **Graph-based test-time scaling**, allowing multi-round query embedding refinement and enabling **million-level ranking**, which prior LLM rankers structurally cannot support. Clarified in **Introduction & Section 3**. |
| **Ablation, Backbone Fairness & Component Contributions** | • “Improvements may stem from Qwen backbone.” (Q9mF) <br> • “Ablation gains seem limited.” | We show that directly using Qwen3 (0.6B/4B) yields much lower performance than LRanker, and that removing global info or test-time scaling decreases accuracy by **11–56%**. All baselines are given **identical 1024-dim offline embeddings**, ensuring fairness. Added to **Appendix E.5**. |
| **Robustness to Extremely Large Candidate Pools** | • “Upper limit? How does performance degrade?” (TeHG, iWYJ) <br> • “Behavior under noisy candidates unclear.” | Using Amazon-23 (4.8M items), we scale candidates from **6.8M → 48M**. LRanker’s performance drops only **~25%** (vs **~50%** for Tiger), following classical **IR scaling-after-saturation** behavior. Latency remains stable because inference depends only on **K, width, depth**, not raw candidate size. Added to **Appendix D**. |
| **Training & Data Splits** | • “Need more details on how data is divided and how training works.” (TeHG, Q9mF) | We expanded Sections 3.1–3.2: <br> • Full description of **leave-one-out** protocol for recommendation tasks <br> • **8:1:1** split for MS MARCO & ESCI <br> • Two-stage pipeline (offline embeddings → clustering → partition sampling → LoRA tuning). Fully clarified in **lines 327–353 and Figure 1**. |
| **Figure 1 Clarity** | • “Test-time scaling is hard to understand.” (Q9mF) | Rebuilt Figure 1 with **step-by-step panels**, explicitly separating (1) training, (2) global aggregation, (3) initial embedding, (4) graph search, (5) final aggregation. Improved readability substantially. |

We believe these updates comprehensively address the reviewers’ concerns and significantly strengthen the contribution of LRanker.

---

### Note · Program_Chairs · 2026-01-17
**Submission Desk Rejected by Program Chairs**

The following references in this submission do not refer to real documents and/or have major errors in bibliographic information:

 Yixin Sun, Yiqun Zhang, Jiaxin Ma, Yanyan Liu, Yanyan Shao, and Shaoping Zhou. Rankgpt: Enhancing zero-shot ranking with instruction-finetuned large language models. arXiv preprint arXiv:2304.09542, 2023b.